# mRNA vaccine developed for sequential selective organ-to-cell targeting of glioma

Lin Shi[1,2], Yueying Li[1,2], Shiyun Huang[1], Jin Peng[1], Chuyao Liu[1], Zuoyu Hu[1], Wenbin Deng [1] ✉ & Guanjun Deng [1] ✉

Messenger RNA (mRNA)-based therapeutics hold great potential for effectively treating various diseases. However, the targeting of mRNA delivered by systemically administered lipid nanoparticles (LNP) is currently limited to the liver. Safe and efficient systemic delivery of mRNA to specific organs and cells remains a major challenge, and it is still unclear whether the positional isomerism of individual compounds within LNP affects their activity. Here, we synthesize a library of meta/ortho/para-ionizable lipidoids and prepare three-component lipid nanoparticles (tLNP) without PEG-lipids. Our findings show that tLNP containing meta-ionizable lipidoids (meta-tLNP) exhibits higher mRNA delivery efficiency than those containing ortho-/para-ionizable lipidoids (ortho-/para-tLNP). Additionally, we report a strategy termed Sequential Selective Organ-to-Cell Targeting (SSOCT), which enables the systemic administration of meta/ortho/para-tLNP to first achieve selective mRNA expression in the spleen, followed by targeted mRNA expression in dendritic cells within the spleen. Notably, we demonstrate that delivering the mRNA vaccine (mXO10 tLNP@mIDH1) using meta-tLNP effectively treats glioma in mice, particularly when combined with Anti-PD-1 therapy. This combination further enhances therapeutic efficacy, even completely eradicating glioma, reducing hepatotoxicity, and minimizing PEG-lipid-induced allergic reactions. This study establishes that mRNA therapy, developed by selectively targeting splenic dendritic cells via SSOCT, represents a promising therapeutic intervention for glioma.

Messenger RNA (mRNA)-based therapies hold revolutionary potential in vaccines, protein replacement therapies, gene editing therapies, and cancer immunotherapies[1]. The U.S. Food and Drug Administration (FDA) has authorized two COVID-19 mRNA vaccines based on lipid nanoparticles (LNP) for emergency use, marking an important milestone for mRNA therapies[2,3]. Beyond COVID-19, mRNA vaccine targeting influenza, herpes viruses, and melanoma have also been developed and entered clinical trials[4–8]. The clinical success of these transformative therapies largely depends on the development of safe, efficient, and highly selective delivery systems that target mRNA to specific organs and cells. Since mRNA primarily accumulates in the liver after LNP systemic delivery, clinical interest has so far mostly focused on liver diseases[9–11]. However, to fully realize the potential of mRNA therapies, there is an urgent need to develop delivery vectors that can specifically target mRNA to extrahepatic tissues. To bypass liver accumulation and achieve organ/cell-specific delivery, scientists have modified the surface of LNP by adding targeting moieties such as peptides or antibodies[12,13]. Recently, a selective organ targeting (SORT) strategy has been developed, which involved incorporating SORT molecules to precisely regulate the in vivo distribution of LNP[14].

---

[1]School of Pharmaceutical Sciences (Shenzhen), Shenzhen Campus of Sun Yat-sen University, Shenzhen, PR China. [2]These authors contributed equally: Lin Shi, Yueying Li. ✉e-mail: dengwb5@mail.sysu.edu.cn; denggj3@mail.sysu.edu.cn

Antibodies against CD117 were conjugated with LNP to develop LNP specifically targeting hematopoietic stem cell[15]. These strategies have demonstrated advantages in reducing liver accumulation and directing mRNA delivery to the extrahepatic tissues or targeting to specific cells[16–18]. Although organ-selective LNP can achieve selective mRNA expression in specific organs, each organ contains different types of cells. Expressing the same mRNA in these different cell types can trigger varied biological functions. Similarly, although cell-targeting LNP achieve targeted mRNA expression in specific cells, the same type of cells is distributed across different parts of the body. The expression of the same mRNA in these cells in various body regions leads to varying biological functions and potential toxicity. Although we made progress in achieving selective mRNA expression in targeted organs or cells, it remained challenging to achieve mRNA expression in specific cell types within targeted organs. Here, we reported a strategy termed Sequential Selective Organ-to-Cell Targeting (SSOCT), which enabled the systematic design of LNP for the precise primary selective mRNA expression in specific organs, followed by secondary targeted mRNA expression in specific cells within these organs. Thus, the SSOCT took advantage of both organ-selective mRNA expression and cell-targeted mRNA expression, allowing the sequential selective expression of mRNA from organs to cells by LNP, thereby enabling the precise expression of mRNA to specific cells within targeted organs.

Positional isomers can alter the pharmacokinetics, safety, and efficacy of small molecules. However, it remains unclear whether positional isomerism of a single compound within multicomponent LNP affects its in vivo activity. Traditional LNP consist of four components: ionizable lipidoids, phospholipids, cholesterol, and PEG-lipids. Ionizable lipidoids are the most abundant and essential component of LNP, carrying a positive charge under specific conditions, which interacts with the negatively charged phosphate backbone of nucleic acids for facilitating the formation of nanoparticle structures[19–21]. Other researchers and we have demonstrated that the lipidoids structure can impact LNP delivery efficacy, stability, and even in vivo targeting[22–24]. However, the impact of positional isomerism of ionizable lipidoids on LNP delivery efficiency remains poorly understood[25–28]. To address this knowledge gap, we synthesized meta-/ortho-/para-positioned ionizable lipidoids using m/o/p-xylylenediamine and aliphatic carbon chains. It was found that the positional isomerism of ionizable lipidoids in multi-component lipid nanoparticles significantly affected mRNA delivery efficiency. Lipid nanoparticles constructed using these positional isomeric ionizable lipidoids achieved precise mRNA expression in dendritic cells within the spleen through the SSOCT strategy. The structure of ionizable lipidoids and their ability to alter charge according to environmental pH affect the stability of LNP during their preparation and delivery, as well as enhance their biocompatibility by influencing interactions with biological systems. We introduced phenyl rings and hydroxyl functional groups into ionizable lipidoids to improve LNP stability and biocompatibility. This approach allowed us to construct a three-component mRNA delivery system (tLNP) without PEG-lipids, thereby avoiding the severe allergic reactions associated with PEG-lipids[29,30]. In summary, we developed a positional isomeric three-component LNP (tLNP) with targeted mRNA expression in DC cells of the spleen through the SSOCT strategy. This approach proved highly advantageous for the development of high-efficiency tumor mRNA vaccines, while also reducing toxicity to organs such as the liver and preventing severe allergic reactions.

Glioma is a type of malignant tumor of the central nervous system characterized by gene mutations, accounting for approximately 80% of primary malignant brain tumors. Currently, the standard clinical treatments for glioma primarily include surgical resection, chemotherapy, and radiotherapy[31,32]. The 2-year survival rate after standard treatment is less than 30%, the 5-year survival rate is less than 10%, and the median survival time is only 12–15 months. Glioma seriously endanger human health[33]. Currently, there are no drugs available that can completely cure glioma. In recent years, immunotherapy has emerged as a cancer treatment by activating the body's immune system to eliminate malignant tumor cells. It has shown high efficacy, high specificity, and low toxicity, marking a significant breakthrough in the field of cancer treatment[34,35]. Increasing clinical research indicates that tumor vaccine is a promising form of tumor immunotherapy, capable of inducing effective anti-tumor immune responses and long-term immune memory against tumors. Traditional tumor vaccine is mainly divided into cellular vaccine, peptide vaccine, and DNA vaccine. However, these traditional tumor vaccines suffer from low safety, low protective efficiency, and long production cycles[36–38]. mRNA vaccine, as a new type of vaccine, has achieved outstanding results in the prevention and control of the COVID-19 pandemic[39]. Compared with traditional vaccines, mRNA vaccine has extremely high protective efficiency, no risk of insertional mutagenesis due to genomic integration, simple design and manufacturing, and the ability to be mass-produced quickly[40,41]. They can generate nearly any tumor-associated antigen, making tumor mRNA vaccine a highly promising treatment for glioma.

In this work, we synthesize a library of meta-/ortho-/para-positioned ionizable lipidoids using m/o/p-xylylenediamine and aliphatic carbon chains. We then prepare three-component lipid nanoparticles (tLNP) using these positional isomeric ionizable lipidoids, phospholipids, and cholesterol for mRNA delivery. We find that meta-tLNP exhibits higher mRNA delivery efficiency than ortho- and para-tLNP in both in vitro and in vivo experiments. In addition, meta-/ortho-/para-tLNP achieve precise mRNA expression in DC cells in the spleen via a sequential selective organ-to-cell targeting (SSOCT) strategy (Fig. 1). Since the spleen is the largest immune organ in the human body, serving as the center of both cellular and humoral immunity, and because DC cells in the spleen are among the most effective antigen-presenting cells, these positional isomeric tLNP are advantageous for enhancing the efficacy of tumor-targeting mRNA, reducing liver toxicity, and minimizing severe allergic reactions caused by PEG-lipids. We further evaluate the ability of mXO10 tLNP to activate an immune response by delivering glioma-associated antigen *IDH1 R132H* mRNA to DC cells in the spleen using the SSOCT strategy. The results show that the mIDH1 mRNA vaccine (mXO10 tLNP@mIDH1) deliver by mXO10 tLNP exhibits strong anti-glioma effects in mice, especially when combined with Anti-PD-1 therapy, resulting in improved efficacy and even complete eradication of glioma. This offers a therapeutic approach for glioma.

## Results
### Synthesis of ionizable lipidoids and library screening

To verify whether the positional isomers of a single compound in multi-component lipid nanoparticles affect their function, we synthesized meta/ortho/para-substituted ionizable lipidoids through the reaction of meta/ortho/para-xylylenediamine with 1,2-epoxy carbon chains. These meta/ortho/para-substituted ionizable lipidoids were named $mXO(n+1)$, $oXO(n+1)$, and $pXO(n+1)$ ($n = 7$, 9, 11, 13, 15) (Fig. 2a). Subsequently, the representative ionizable lipidoids m/o/pXO10 were further purified and characterized by $^1H$ nuclear magnetic resonance spectroscopy ($^1H$ NMR). (Figs. S1–S3). We incorporated phenyl rings and hydroxyl functional groups into ionizable lipidoids to enhance the stability and biocompatibility of LNP. This approach enabled us to construct three-component lipid nanoparticles (tLNP) without PEG-lipids, avoiding the severe allergic reactions associated with PEG-lipids. Then, $mXO(n+1)$, $oXO(n+1)$, and $pXO(n+1)$ were respectively mixed with DOPE and cholesterol in a molar ratio of 9:6:4 to prepare the three-component lipid nanoparticles (meta-, ortho-, and para-tLNP), which were then mixed with mRNA in specific ratios to produce mRNA-loaded tLNP (meta-/ortho-/para-tLNP@mRNA). We then prepared these meta-/ortho-/para-tLNP@mLuc by delivering

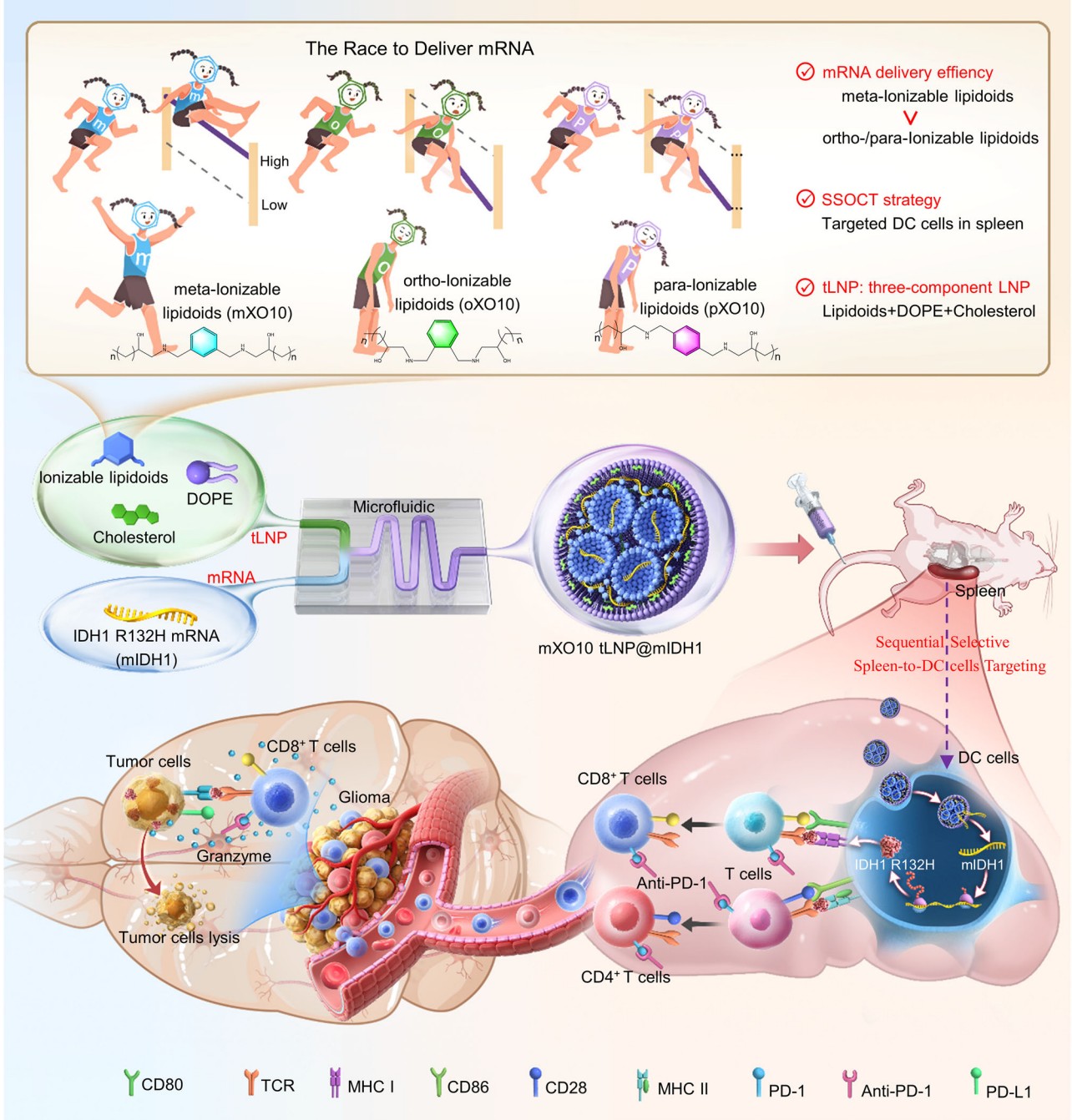

**Fig. 1 | Schematic illustration of the preparation and screening of position isomerized three-component LNP (tLNP) and tLNP achieved targeted mRNA expression in DC cells within the spleen through the SSOCT strategy and the schematic of mXO10 tLNP-delivered mRNA vaccine combined with Anti-PD-1 in the treatment of glioma.** We synthesized a library of ionizable lipidoids with meta-, ortho-, or para-substitutions using the corresponding phenylenediamine isomers and aliphatic carbon chains. These positionally isomeric ionizable lipidoids were then used, along with phospholipids and cholesterol, to formulate three-component lipid nanoparticles (tLNP) for mRNA delivery. Among them, the meta-tLNP demonstrated higher mRNA delivery efficiency compared to ortho-/para-tLNP. Furthermore, tLNP enabled precise mRNA expression in splenic dendritic cells via a sequential selective organ-to-cell targeting (SSOCT) strategy. Notably, the mXO10 tLNP-delivered mIDH1 mRNA vaccine (mXO10 tLNP@mIDH1) showed potent anti-glioma activity, which was further enhanced when combined with anti-PD-1 therapy.

firefly luciferase mRNA (mLuc) with meta-/ortho-/para-tLNP, and evaluated the mRNA delivery efficiency of tLNP in cells by measuring luciferase expression. Interestingly, the meta-tLNP showed significantly higher mRNA delivery efficiency than the ortho- and para-tLNP (Fig. 2b). The mXO10 tLNP delivering mRNA at 1.6 times the efficiency of lipo2000 and 1.4 times that of the commercial ALC-0315 LNP. These results indicated that positional isomers of tLNP could influence mRNA delivery efficiency, with meta-tLNP delivering mRNA more effectively than ortho-/para-tLNP. To verify whether the type of group on the tail chain eliminated the effect of positional isomers of ionizable lipidoids on mRNA delivery efficiency, we synthesized meta/ortho/para-substituted ionizable lipidoids through a Michael addition reaction of meta-/ortho-/para-xylylenediamine with carbon chains containing acrylate groups. These positional isomers were named mXE(n + 3), oXE(n + 3), and pXE(n + 3) (n = 7, 9, 11, 13, 15). Subsequently, the representative ionizable lipidoids m/o/pXE14 was further

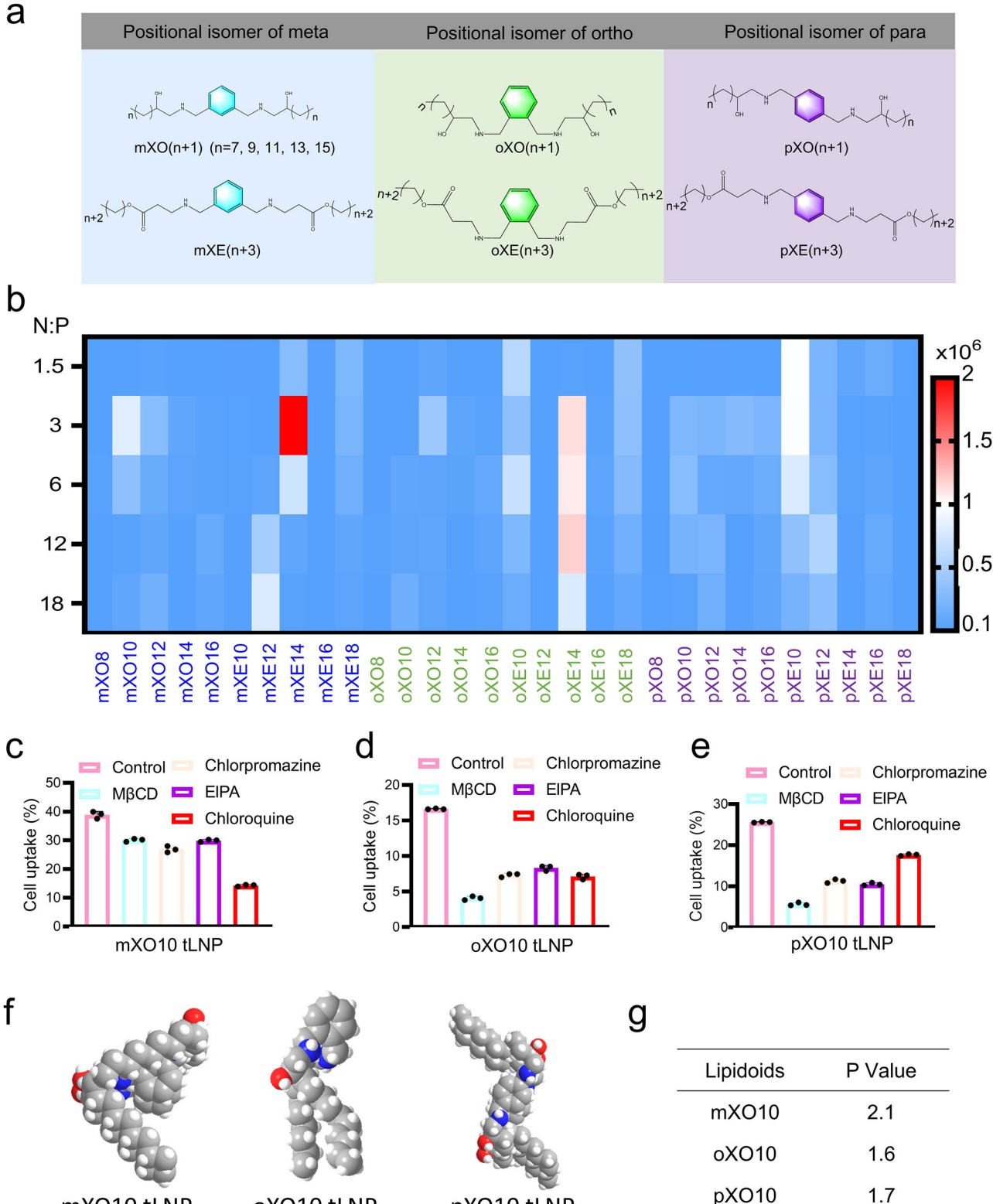

**Fig. 2 | Synthesis and screening of the position-isomerized lipidoids library.** **a** Molecular structures of meta-, ortho- and para-ionizable lipidoids. **b** Heat map showing in HEK-293 cells transfection data after treatment with tLNP made from 30 types of lipidoids encapsulating mLuc. Study on the cellular uptake pathways of mRNA delivered by mXO10 tLNP (**c**), oXO10 tLNP (**d**), and pXO10 tLNP (**e**). (n = 3 independent experiments, data were presented as the mean ± SEM) **f** 3D structures of mXO10, oXO10, and pXO10 lipidoids. **g** Calculated parameters and P values of mXO10, oXO10, and pXO10 lipidoids. Source data are provided as a Source data file.

purified and characterized by $^1$H NMR (Figs. S4–S6). The mXE(n + 3), oXE(n + 3) and pXE(n + 3) were respectively mixed with DOPE and cholesterol in a 9:6:4 molar ratio to prepare meta-/ortho-/para-tLNP. The results showed that the meta-tLNP delivered mRNA more efficiently than the ortho-/para-tLNP, with the mXE14 tLNP delivering mRNA at 4 times the efficiency of lipo2000 and 3.6 times that of the ALC-0315 LNP. These findings suggested that although the type of functional group on the tail carbon chain might have affected the mRNA delivery efficiency of tLNP, it did not alter the conclusion that meta-tLNP delivered mRNA more efficiently than ortho-/para-tLNP, further confirming the previous conclusions.

Cell membrane and lysosome escape are key barriers to intracellular mRNA delivery[42]. The positional isomers of tLNP resulted in different mRNA delivery efficiencies, prompting us to investigate their cellular uptake and endosomal escape characteristics. Confocal laser scanning microscopy (CLSM) images showed no significant difference in lysosome escape among meta-, ortho- and para-tLNP (Figs. S7–S11). However, the cellular uptake of meta-tLNP was significantly higher than that of ortho-/para-tLNP, suggesting that the higher mRNA delivery efficiency of meta-tLNP was due to their superior cellular uptake (Fig. S12). Next, we studied the pathway of cellular uptake of tLNP by using inhibitors to block different cellular uptake pathways of nanoparticles. These results showed that the cellular uptake efficiency of mXO10 tLNP decreased by 63% after the addition of chloroquine, which altered the pH on the cell membrane surface and inhibited endocytosis[43]. This indicated that endocytosis was the main pathway for the cellular uptake of mXO10 tLNP (Fig. 2c). On the other hand, MβCD reduced the cellular uptake efficiency of oXO10 tLNP and pXO10 tLNP by 75% and 78%, respectively, suggesting that the primary pathway for their uptake by cells was clathrin-dependent endocytosis (Fig. 2d, e). Therefore, meta-tLNP were more efficient in delivering mRNA than ortho-/para-tLNP due to their higher cellular uptake. To understand why positional isomers lead to differences in the cellular uptake of tLNP, we conducted an in-depth study on the relationship between the geometry of ionizable lipidoids and their mRNA delivery efficiency. We calculated the P values of three ionizable lipidoids (mXO10, oXO10, and pXO10). As an important parameter, the P value can provide key clues for us to predict the nanostructures that ionizable lipidoids may form under specific conditions. $P = v/(\ell a)$, a represents the head area of the ionizable lipidoids molecule, which reflects the size and shape of the lipidoids molecule head; v and $\ell$ represent the tail volume and length of the ionizable lipidoids molecule, respectively. These physical properties of the tail have an important influence on the overall structure and function of the lipidoid molecule. We can calculate the P value of different ionizable lipidoids to predict the nanostructures they may form. When the P value is less than 1/3, these ionizable lipidoids molecules tend to form spherical structures during aggregation. This structure has relatively stable physical and chemical properties, but it is not as conducive to vector delivery as other structures; when $1/3 \leq P \leq 1/2$, ionizable lipidoids will be arranged into hexagonal phase (HI phase) nanostructures; and when $1/2 < P \leq 1$, planar nanostructures will be formed. Hexagonal phase (HI phase) and planar nanostructures have different characteristics and mechanisms of action in the process of lipid nanoparticle delivery. Compared with spherical structures, they may be more conducive to vector delivery in some aspects. When P is greater than 1, this type of ionizable lipidoids has the typical structural characteristics of "small head and large tail". This special structure makes them easier to form HII phase or reverse spherical nanostructures, and the larger the P value, the easier it is to form this structure. The interaction between HII phase and reverse spherical nanostructure and endosomal membrane in biological function effectively improves the delivery efficiency of carriers. After calculation, we found that the P value of mXO10 was 2.1, which was significantly higher than the P value of oXO10 (P = 1.6) and the P value of pXO10 (P = 1.7) (Fig. 2f, g). mXO10 was more likely to assemble into

HII phase nanostructures, which provided more favorable conditions for intracellular delivery of mRNA. These experimental results indicated that, compared with oXO10 and pXO10, mXO10 was more readily assembled into HII phase nanostructures, leading to higher cellular uptake efficiency of meta-tLNP and consequently resulting in improved mRNA delivery efficiency. Thus, we selected the meta-tLNP (mXO10 tLNP and mXE14 tLNP) with the best mRNA delivery efficiency for further experiments. In the following text, mXO10 tLNP@mRNA/mXE14 tLNP@mRNA refered to the three-component lipid nanoparticles prepared using mXO10/mXE14, DOPE, and cholesterol in a 9:6:4 molar ratio, with mRNA-loaded nanoparticles prepared by microfluidics.

## Discovery and development of SSOCT

To study the systemic mRNA delivery characteristics of meta-/ortho-/para-tLNP in vivo, we encapsulated mRNA encoding firefly luciferase (mLuc) in m/o/pXO10 tLNP and m/o/pXE14 tLNP, a reporter protein that can be visualized in vivo using the IVIS imaging system. At a dose of 0.5 mg/kg, administered via tail vein injection in mice, luciferase expression was observed at 6 h, with commercial ALC-0315 four-component LNP as the positive group. Interestingly, the bioluminescent signal of mice treated with m/o/pXO10 tLNP and m/o/pXE14 tLNP was only present in the spleen (Figs. 3a–c, S13 and S14a). In contrast, ALC-0315@mLuc showed strong protein expression levels in both the liver and spleen. The result displayed that by altering the structure of ionizable lipidoids, liver-extrinsic organ (e.g., spleen)-selective expression can be achieved with LNP. To investigate the effect of dose on tLNP targeting, we designed three different dosing regimens for mXO10 and ALC-0315: 2 mg/kg, 1.5 mg/kg, 1 mg/kg, 0.3 mg/kg, 0.2 mg/kg, and 0.1 mg/kg. Six hours after administration, bioluminescence imaging analysis was performed. The results showed that whether it was the high dose of 2 mg/kg or the low dose of 0.1 mg/kg, the mXO10 tLNP could selectively express mRNA in the spleen of mice. The targeted delivery properties of tLNP remained unchanged despite the variation in dose (Figs. S15 and S16). SSOCT demonstrated organ-selective mRNA expression in the first phase through m/o/pXO10 tLNP and m/o/pXE14 tLNP, effectively delivering mRNA to the spleen. In the second phase, SSOCT facilitated targeting mRNA expression in specific cells within the targeted organs, achieving precisely controlled mRNA expression to these specific cells. tLNP with SSOCT selectively expressed mRNA in target organs without causing mRNA expression in other parts of the body, thereby enhancing the targeted delivery of mRNA by tLNP. In systemic mRNA delivery studies in vivo, mXO10 tLNP showed selective mRNA expression in the spleen via the SSOCT strategy. To further examine the cell-targeted mRNA expression of mXO10 tLNP in the spleen, we used Ai9 mice and administered mXO10 tLNP loaded with Cre mRNA (mCre) at two different doses (0.5 mg/kg and 0.2 mg/kg). Spleen tissues were collected and analyzed via flow cytometry to detect tdTomato-positive cell types, including B cells, T cells, DC cells and NK cells (NK1.1$^+$). The results showed that, compared with the control group, a significant increase in tdTomato expression was observed only in DC cells following treatment with mXO10 tLNP@mCre. Furthermore, the level of tdTomato expression in DC cells was higher than that observed in the other three cell types (B cells, T cells, and NK cells). This suggested that mXO10 tLNP@mCre secondarily targeted DC cells in the spleen to enhance tdTomato expression (Fig. 3d). Notably, this cell-targeting was consistent across both high- and low-dose groups, indicating that the observed cell-targeting was independent of the administration dose (Fig. S17). The SSOCT strategy refers to achieving organ-selective mRNA expression in the first phase, followed by cell-targeted expression within that organ in the second phase. This enables precise enhancement of mRNA expression in defined cell populations within a target organ. In this case, mXO10 tLNP@mRNA selectively expressed mRNA in the spleen, with no detectable expression in other organs, and

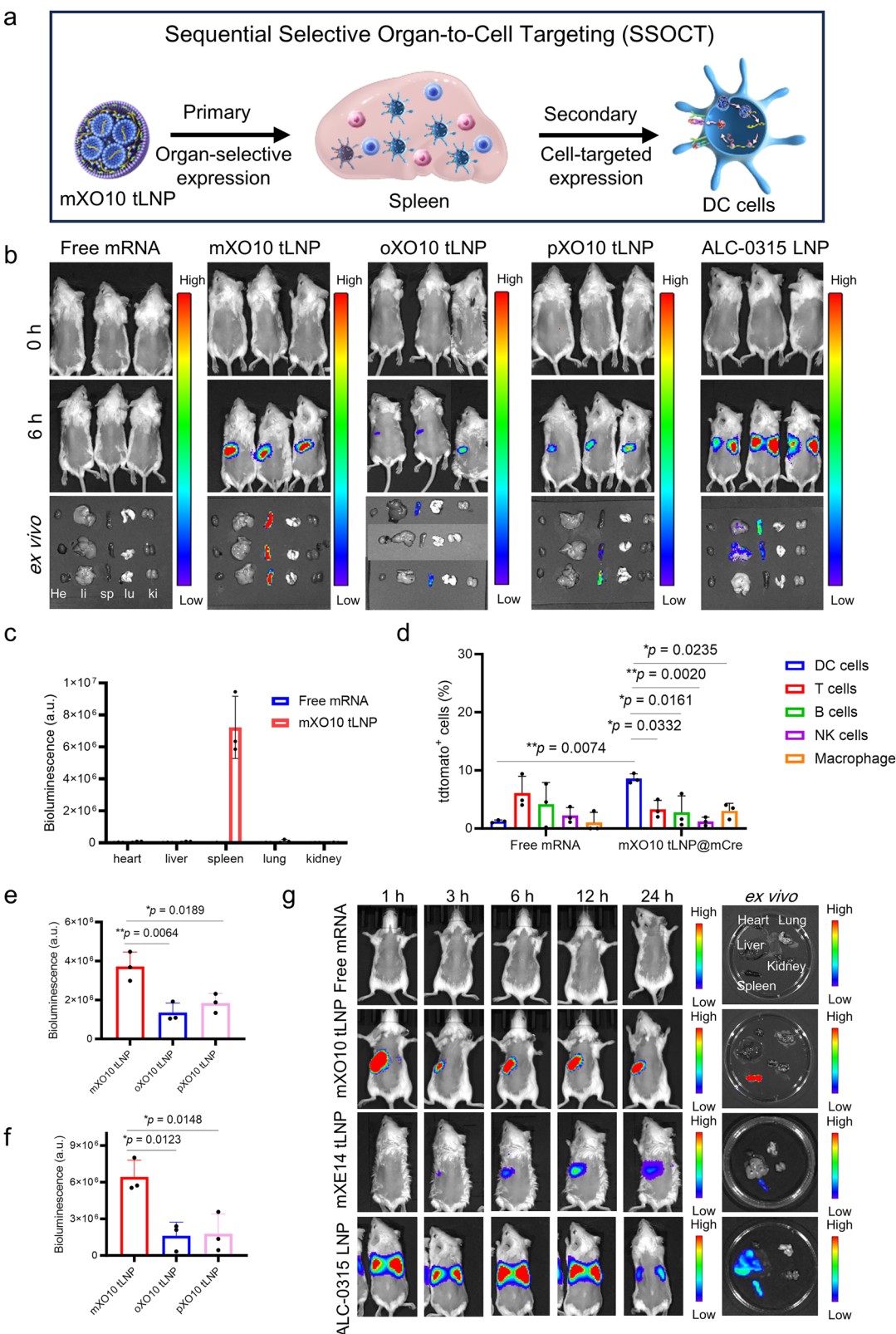

subsequently facilitated targeted expression in DC cells within the spleen. Thus, tLNP utilizing the SSOCT approach enabled precise enhancement of mRNA expression in DC populations within the spleen while minimizing off-target expression. There are multiple subtypes of DC in the spleen—cDC1, cDC2, and pDC—each playing distinct roles in immune function. We further employed flow cytometry to investigate which DC subpopulation was targeted by mXO10 tLNP for mRNA

expression. The experimental results showed that, compared with the control group, there was no significant difference in the expression of mXO10 tLNP@tdTomato mRNA in cDC1 and pDC. However, a significant increase in expression was observed in cDC2, indicating that mXO10 tLNP@mRNA specifically targeted cDC2 to enhance mRNA expression levels (Fig. S18). The spleen is a key organ of the human immune system and the center of both cellular and humoral immunity.

**Fig. 3 | Study on the effect of positional isomers on mRNA expression in vivo and the development of SSOCT. a** Schematic diagram of SSOCT strategy, mXO10 tLNP first targeted the spleen in mice, and then precisely targeted DC cells in the spleen. **b** Study on in vivo delivery of mRNA by mXO10 tLNP, oXO10 tLNP, and pXO10 tLNP. **c** Semi-quantification of ex vivo organ images (heart, liver, spleen, lungs, kidneys) of mXO10 tLNP group and free mRNA group in the Fig. 2b. (n = 3 independent experiments, data were presented as the mean ± SEM) **d** Study on targeted delivery of mRNA to cells in the spleen using mX10 tLNP at a dose of 0.5 mg/kg. (n = 3 independent experiments, data were presented as the mean ± SEM, and analyzed using two-way ANOVA followed by Tukey post hoc test, 95%

confidence interval.) **e** Semi-quantitative analysis of in vivo spleen images at 6 h in Fig. 2b. (n = 3 independent experiments, data were presented as the mean ± SEM, and analyzed using one-way ANOVA followed by Tukey post hoc test, 95% confidence interval.) **f** Semi-quantitative analysis of the ex vivo spleen images in Fig. 2b. (n = 3 independent experiments, data were presented as the mean ± SEM, and analyzed using one-way ANOVA followed by Tukey post hoc test, 95% confidence interval., *$p < 0.05$, **$p < 0.01$, ***$p < 0.001$, ****$p < 0.0001$, n.s. not significant.) **g** The in vivo translation of mRNA delivered by mXO10 tLNP at different time points into proteins was investigated. Source data and exact $p$ values are provided as a Source data file.

DC cells in the spleen are among the most effective antigen-presenting cells in the body. Therefore, an mRNA delivery system that selectively targets DC cells in the spleen through the SSOCT strategy can enhance the therapeutic efficacy of mRNA vaccines. The free mRNA group did not show protein expression in mice, as mRNA was easily degraded. In contrast, the m/o/pXO10 tLNP and m/o/pXE14 tLNP groups exhibited strong protein expression levels. These results demonstrated that tLNP significantly enhanced mRNA stability and the protein translation capability of mRNA in vivo. Notably, we found that the positional isomers of ionizable lipidoids affected the in vivo mRNA delivery efficiency of tLNP. After intravenous injection of m/o/pXO10 tLNP@mLuc in mice, the in vivo bioluminescence intensity of the mXO10 tLNP group was the highest in all group, showing the highest mRNA delivery of mXO10 tLNP (Fig. 3e, f). The result demonstrated that meta-tLNP delivered mRNA in vivo significantly more efficiently than ortho- and para-tLNP, and these findings were consistent with the in vitro results. The experiment revealed that although the bioluminescence signal in mice treated with m/o/pXE14 tLNP@mLuc was weak, ex vivo imaging showed relatively strong bioluminescence signal localized in the spleen. Additionally, the in vivo bioluminescence signal of the mXE14 tLNP@mLuc group was significantly higher than that of the oXE14 and pXE14 tLNP@mLuc groups, further demonstrating that meta-tLNP was more effective than ortho- and para-tLNP in mRNA delivery (Fig. S14b, c). We also found that the mRNA delivery efficiency of m/o/pXO10 tLNP in vivo was significantly higher than that of m/o/pXE14 tLNP, while in vitro, the mRNA delivery efficiency of m/o/pXO10 tLNP was lower than that of m/o/pXE14 tLNP (Figs. 2b, 3e and S14b). This suggests that although tLNP efficiently delivered mRNA into cells in vitro, it did not necessarily ensure efficient mRNA delivery in vivo. We further investigated the in vivo translation of mRNA delivered by mXO10 tLNP and mXE14 tLNP into protein at different time points. The results showed that mXO10 tLNP and mXE14 tLNP delivered mRNA that expressed protein for at least 24 h and exhibited a pulsatile pattern of protein expression (Fig. 3g). Next, we studied the biodistribution of mXO10 tLNP@mLuc and mXE14 tLNP@mLuc in vivo, and the results showed that, compared to the free mRNA group, mXO10 tLNP and mXE14 tLNP were distributed throughout the body at 1 h and primarily accumulated in the liver, kidneys, and spleen after 24 h. In contrast, no fluorescence signals were detected in the control group mice, indicating that the free mRNA was rapidly degraded (Fig. S19). These findings suggested that tLNP effectively enhanced mRNA stability and organ-targeted distribution. These results also showed that although tLNP had good biodistribution in major organs of mice, only some organs effectively translated mRNA into protein. Therefore, biodistribution alone was not sufficient to achieve functional mRNA delivery, and the key lay in the specific biological effects of the lipid nanoparticles on organ tissues and cells.

## Characterization of tLNP@mRNA

We optimized the ratio of tLNP/mRNA to achieve the highest mRNA delivery efficiency. The intracellular bioluminescence intensity increased as the molar ratio of nitrogen in mXO10 tLNP to phosphorus in mRNA (mLuc) increased to 3:1. (The bioluminescence signal in cells was strongest when the molar ratio of nitrogen in mXE14 tLNP to

phosphorus in mLuc was 6:1.) Ultimately, an N-to-P ratio of 3:1 was chosen to prepare mXO10 tLNP@mRNA for subsequent studies (and an N-to-P ratio of 6:1 was selected to prepare mXE14 tLNP@mRNA) (Fig. 4a). We then characterized the physical and chemical properties of mXO10 tLNP@mRNA and mXE14 tLNP@mRNA. Dynamic light scattering instruments were used to measure the hydrated particle size of tLNP. The results showed that the average particle size of both mXO10 tLNP@mRNA and mXE14 tLNP@mRNA was approximately 180 nm (Fig. 4b). We also measured the Zeta potential of tLNP, finding that the surface potential of mXO10 tLNP@mRNA was approximately −15.04 mV, while that of mXE14 tLNP was −17.95 mV. Transmission electron microscopy (TEM) images showed that both mXO10 tLNP@mRNA and mXE14 tLNP@mRNA had a dense, spherical structure, with nanoparticle sizes around 170 nm (Fig. 4c). The mRNA encapsulation efficiency for both mXO10 tLNP and mXE14 tLNP exceeded 70% (Fig. 4d). The pKa of tLNP was determined using 6-(p-toluidino)-2-naphthalenesulfonic acid (TNS) assays, indicating pKa values of 4.45 and 5.70 for mXO10 tLNP@mRNA and mXE14 tLNP@mRNA, respectively (Fig. 4e). These values were conducive to the delivery and release of mRNA by the tLNP in vivo. As shown in Fig. S20a, b, the particle sizes of mXO10 tLNP@mRNA and mXE14 tLNP@mRNA exhibited no significant changes over the course of one week, suggesting that the tLNP maintained their stability during this period. Agarose gel electrophoresis was used to evaluate the encapsulation efficiency of tLNP for mRNA. Compared to naked mRNA, it was observed that mRNA delivered by mXO10 tLNP and mXE14 tLNP was not degraded, even in the presence of RNase (200 ng/μL). These results indicated that mXO10 tLNP and mXE14 tLNP effectively loaded mRNA and significantly enhanced its stability and resistance to RNase degradation (Fig. S20c). In vitro hemolysis experiments revealed that neither mXO10 tLNP@mRNA nor mXE14 tLNP@mRNA caused hemolysis, demonstrating their good biocompatibility (Fig. S21). The results of CCK-8 assay showed that tLNP prepared with ionizable lipidoids mXO10 and mXE14 exhibited almost no cytotoxicity, even at ionizable lipidiods concentrations up to 50 μM (Fig. S22). Therefore, tLNP prepared from these ionizable lipidoids have efficient mRNA delivery and excellent biocompatibility.

## Study on cellular uptake and delivery of different mRNA by tLNP

Cellular uptake and endosomal escape of lipid nanoparticles are critical for the intracellular delivery of mRNA. We next investigated the efficiency of mRNA delivery by mXO10 tLNP and mXE14 tLNP in cellular uptake. Cy5-labeled firefly luciferase mRNA (Cy5-mLuc) was delivered to DC2.4 and GL261 cells using mXO10 tLNP and mXE14 tLNP. CLSM images showed that both mXO10 tLNP and mXE14 tLNP effectively delivered mLuc into DC2.4 and GL261 cells compared to the control group (Figs. 4f and S23). Flow cytometry results indicated that mXO10 tLNP delivered mLuc to DC2.4 cells with 4 times the efficiency of the free mRNA group, while mXE14 tLNP was 7.7 times more efficient, demonstrating that tLNP significantly enhanced mRNA uptake and intracellular delivery efficiency (Figs. 4g and S24). The acidic environment of lysosomes protonates the ionizable lipidiods nanoparticles, allowing them to escape from lysosomes and ensure that the mRNA reaches the cytoplasm. Therefore, lysosomal escape of LNP is a

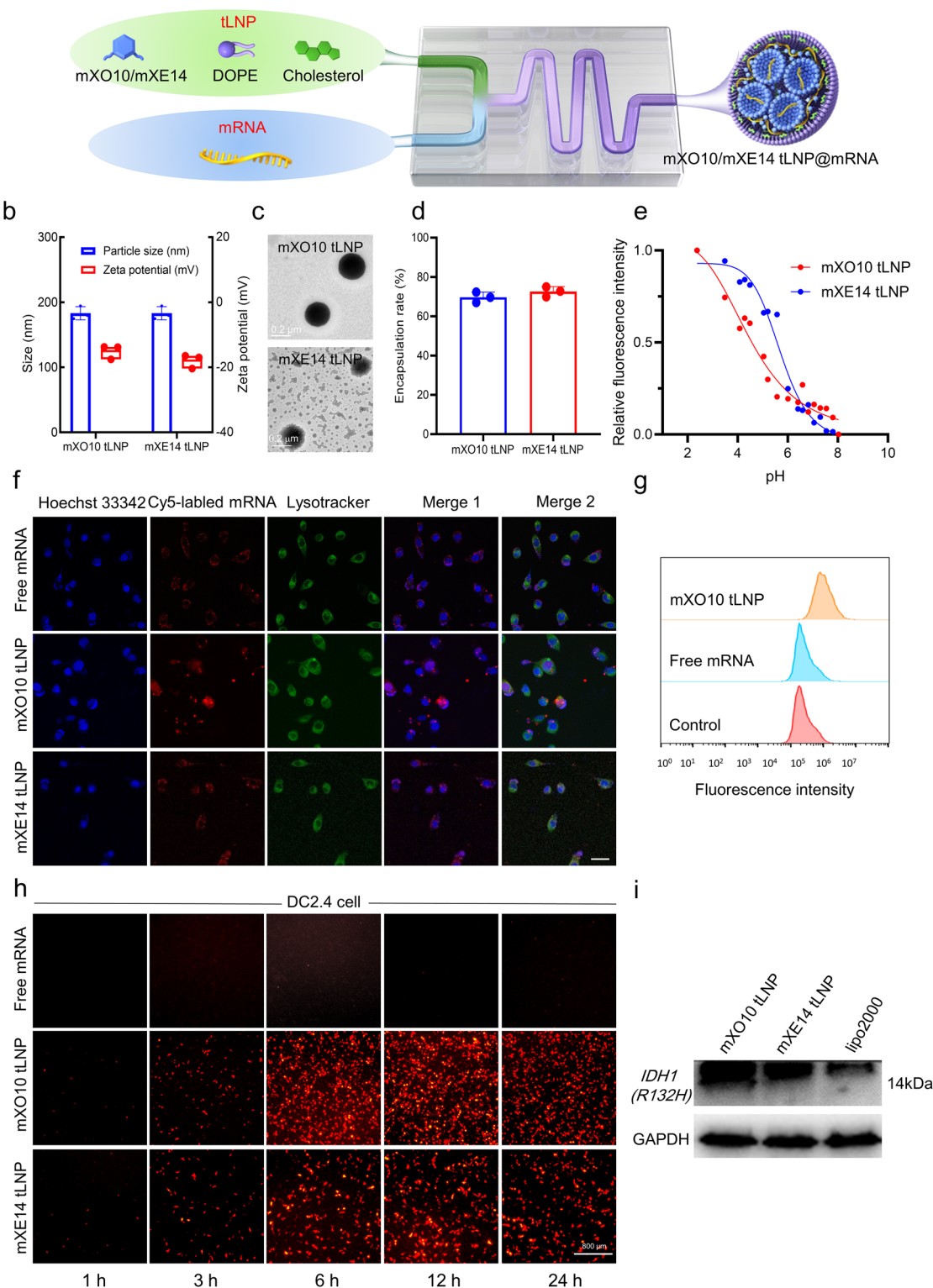

critical factor in mRNA delivery. We studied the lysosomal escape of tLNP using CLSM. The result showed that, compared to the control group, naked mRNA could barely enter DC2.4 cells, whereas mXO10 tLNP@mRNA and mXE14 tLNP@mRNA showed significant lysosomal escape and a substantial amount of mRNA entering the cytoplasm, providing a foundation for efficient protein expression. The result indicated that mXO10 tLNP and mXE14 tLNP significantly improved the

efficiency of mRNA uptake and lysosomal escape, enhancing the delivery and translation of mRNA into proteins. Previous data have shown that mXO10 tLNP and mXE14 tLNP effectively deliver luciferase mRNA (mLuc) into cells and efficiently translate it into luciferase. Next, we explored whether these tLNP could deliver other mRNA and efficiently translate them into proteins. mCherry mRNA (mCherry) encodes a red fluorescent protein. We used mXO10 tLNP and mXE14

**Fig. 4 | Characterization of mXO10 tLNP and mXE14 tLNP and their efficiency in delivering different mRNA to cells. a** Schematic diagram of the preparation process for tLNP loaded with mRNA. **b** Study on the hydrated particle size and Zeta potential of mXO10 tLNP and mXE14 tLNP. (n = 3 independent experiments, data were presented as the mean ± SEM) **c** Representative TEM images of mXO10 tLNP and mXE14 tLNP. Scale bar: 0.2 μm. **d** Study on mRNA encapsulation efficiency of mXO10 tLNP and mXE14 tLNP. (n = 3 independent experiments, data were presented as the mean ± SEM) **e** Study on the pKa values of mXO10 tLNP and mXE14 tLNP. **f** CLSM was used to study the cellular uptake and lysosomal escape of mRNA delivered by mXO10 tLNP and mXE14 tLNP in DC2.4 cells. The mRNA was labeled with Cy5 (Red), lysosomes were stained with Lysotracker (Green), and nuclei were stained with Hoechst 33342 (Blue). The control group consisted of free mRNA labeled with Cy5. Scale bar 20 μm. (n = 3 independent experiments) **g** Flow cytometry to evaluate the efficiency of mRNA delivery by mXO10 tLNP into DC2.4 cells. **h** The expression of mCherry mRNA delivered by mXO10 tLNP and mXE14 tLNP in DC2.4 cells. Scale bar 800 μm. (n = 3 independent experiments) **i** Western blotting analysis of the protein translation of IDH1R132H mRNA (glioma-associated antigen) delivered by mXO10 tLNP (or mXE14 tLNP). (n = 3 independent experiments) Source data are provided as a Source data file.

tLNP to deliver mCherry mRNA into cells and observed the expression of mCherry protein in DC2.4 and GL261 cells at 3 h, 6 h, 18 h, 24 h, and 36 h using CLSM. The results showed very strong red fluorescent signals in both DC2.4 and GL261 cells, with the intensity of the red fluorescence increasing over time, peaking at around 24 h (Figs. 4h and S25). The result demonstrated that mXO10 tLNP and mXE14 tLNP effectively delivered mCherry mRNA into cells and efficiently expressed mCherry protein. Additionally, we investigated whether mXO10 tLNP and mXE14 tLNP could efficiently deliver glioma-associated antigen IDH1R132H mRNA (mIDH1). Western blotting results showed that mXO10 tLNP@mIDH1 and mXE14 tLNP@mIDH1 successfully expressed the *IDH1 R132H* peptide in cells (Fig. 4i). These findings indicated that mXO10 tLNP and mXE14 tLNP were capable of effectively delivering different functional mRNA and translating them efficiently into protein. Therefore, tLNP could serve as a versatile mRNA delivery platform for developing mRNA therapies for various diseases.

## Study on glioma mRNA vaccine (mXO10 tLNP@mIDH1) and its combination with PD-1 antibody (Anti-PD-1) for treating glioma

Previous data showed that both mXO10 tLNP and mXE14 tLNP achieved targeted mRNA expression in DC cells of the spleen via the SSOCT strategy. However, mXE14 tLNP had a lower efficiency in delivering mRNA in vivo. We ultimately chose mXO10 tLNP for further in vivo mRNA delivery studies. The spleen is an important organ in the immune system, involved in immune surveillance, antigen presentation, and lymphocyte activation. DC cells in the spleen are among the most effective antigen-presenting cells in the body. Therefore, mXO10 tLNP was suitable for developing tumor mRNA vaccines, as it could enhance the therapeutic efficacy of tumor mRNA vaccines while reducing toxicity to major organs like the liver. The IDH1 gene encoded isocitrate dehydrogenase 1, which converted isocitrate to alpha-ketoglutarate (α-KG). However, the mutated *IDH1 R132H* enzyme produced the metabolite D-2-hydroxyglutarate (2-HG), which competitively inhibited many α-KG-dependent dioxygenases, leading to histone dysregulation and abnormal angiogenesis, and ultimately inducing glioma. Therefore, the *IDH1 R132H* gene mutation was a specific mutation in glioma and was a driver of glioma development[44]. It was often used as a target antigen for developing glioma vaccines. Next, we used mXO10 tLNP to deliver *IDH1 R132H* mRNA (mIDH1) to prepare a glioma mRNA vaccine (mXO10 tLNP@mIDH1). We then evaluated the efficacy of the mXO10 tLNP@mIDH1 mRNA vaccine and its combination with Anti-PD-1 treatment for glioma (Fig. 5a). First, we established orthotopic brain glioma model in mice using GL261 cells labeled with firefly luciferase. The GL261 cells were also genetically modified to express the *IDH1 R132H* (Fig. S26). The growth of glioma cells was measured via bioluminescence intensity. We found that the bioluminescence intensity was significantly lower in the mXO10 tLNP@mIDH1 mRNA vaccine group compared to the control group (Fig. 5b, c). By day 20, approximately 2/5 of the mice in the mXO10 tLNP@mIDH1 mRNA vaccine group had no detectable bioluminescence signals in the brain, indicating effective tumor growth inhibition. Notably, mice treated with mXO10 tLNP@mIDH1 mRNA vaccine combined with Anti-PD-1 showed the lowest bioluminescence intensity

among all groups, and by day 20, all mice in this treatment group had no observable bioluminescence signals, suggesting that glioma cells were almost entirely eliminated. We then assessed mice survival rates, revealing that the survival rate of mice in the mXO10 tLNP@mIDH1 vaccine group was 60%, significantly higher than the control group (0%) (Fig. 5d). Particularly, the survival rate in the mXO10 tLNP@mIDH1 vaccine combined with Anti-PD-1 treatment group reached 80%. These results suggested that the mXO10 tLNP@mIDH1 mRNA vaccine effectively inhibited glioma growth, and the combination with Anti-PD-1 treatment provided even stronger inhibition. Mouse body weight was an important indicator of tolerance to tumor therapy. The body weight of mice in the mXO10 tLNP@mIDH1 mRNA vaccine group, as well as the group that received the combined mRNA vaccine and Anti-PD-1 treatment, remained relatively stable, while the body weight of the control group mice significantly decreased over time (Fig. 5e). These results suggested that the mRNA vaccine therapy and its combination with Anti-PD-1 treatment were non-toxic to the mice. Further immunohistochemical analysis showed that the level of cell apoptosis was higher in the mXO10 tLNP@mIDH1 mRNA vaccine group than in the control group. The combination of mXO10 tLNP@mIDH1 mRNA vaccine with Anti-PD-1 treatment resulted in the highest degree of apoptosis among all groups (Fig. 5f). The result demonstrated that mXO10 tLNP@mIDH1 and its combination with Anti-PD-1 effectively inhibited the growth of glioma cells. Therefore, these findings suggested that the mXO10 tLNP@mIDH1 mRNA vaccine was an efficient and low-toxicity therapy for glioma, with the combination of the mRNA vaccine and Anti-PD-1 showing particularly strong anti-glioma effects.

## Study on the mechanism of glioma mRNA vaccine (mXO10 tLNP@mIDH1) therapy

Given that the mXO10 tLNP@mIDH1 mRNA vaccine induced a strong immune response against glioma, especially when combined with Anti-PD-1, we aimed to further investigate the underlying mechanism of the mXO10 tLNP@mIDH1 mRNA vaccine and its combination with Anti-PD-1 in glioma immunotherapy. DC cells are the most potent antigen-presenting cells in the body, and their activation is crucial for the antitumor immune response elicited by mRNA vaccines. We analyzed the activation of DC cells in the spleen of mice using flow cytometry. Figure 6a, b showed that, compared to the control group, the activation of DC cells in the mXO10 tLNP@mIDH1 mRNA vaccine group increased significantly, being 12 times higher than in the control group. Notably, the activation of DC cells was highest in the group receiving the mXO10 tLNP@mIDH1 mRNA vaccine combined with Anti-PD-1, reaching 1.5 times that of the mRNA vaccine alone and 17 times that of the control group. Subsequently, we measured the levels of pro-inflammatory cytokines (IL-6, IL-12, and TNF-α) in vivo using ELISA. The results showed that the levels of pro-inflammatory cytokines in the mXO10 tLNP@mIDH1 vaccine group were significantly higher than those in the control group, with the highest levels observed in the group receiving the combination therapy (Figs. 6c and S27). Thus, the mXO10@mIDH1 vaccine effectively stimulated the activation and maturation of DC cells, and when combined with Anti-PD-1, it further enhanced the activation of DC cells and induced the secretion of large

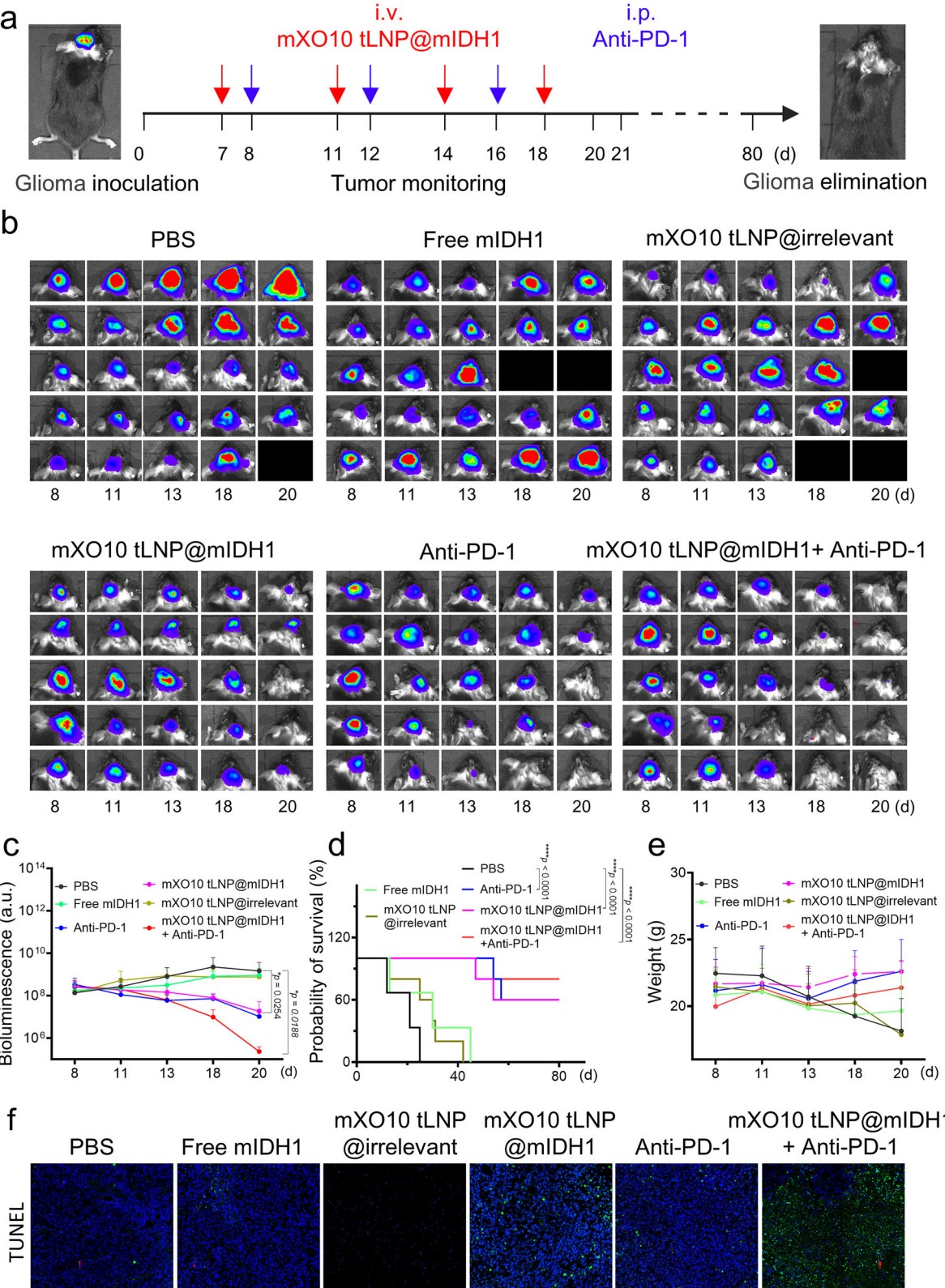

**Fig. 5 | Study on the treatment of glioma using mRNA vaccine combined with Anti-PD-1 therapy. a** Schematic diagram of the treatment process for glioma using the mXO10 tLNP@mIDH1 vaccine combined with Anti-PD-1. **b** Bioluminescence imaging study to monitor glioma growth in vivo. **c** Semi-quantitative analysis of bioluminescence signals in glioma from (**b**) after different treatments. **d** Study on the survival time of mice after different treatments (PBS vs mXO10 tLNP@mIDH1,

****$p$ <0.0001; PBS vs mXO10 tLNP@mIDH1+Anti-PD-1, ****$p$ <0.0001.). **e** Study on the monitoring of mouse body weight during treatment. (n = 5 independent experiments, data were presented as mean ± SEM) **f** Study on apoptosis of glioma cells in mice after different treatments. (n = 3 independent experiments, data were presented as mean ± SEM.) Scale bar 100 μm. (n = 3 independent experiments) Source data are provided as a Source data file.

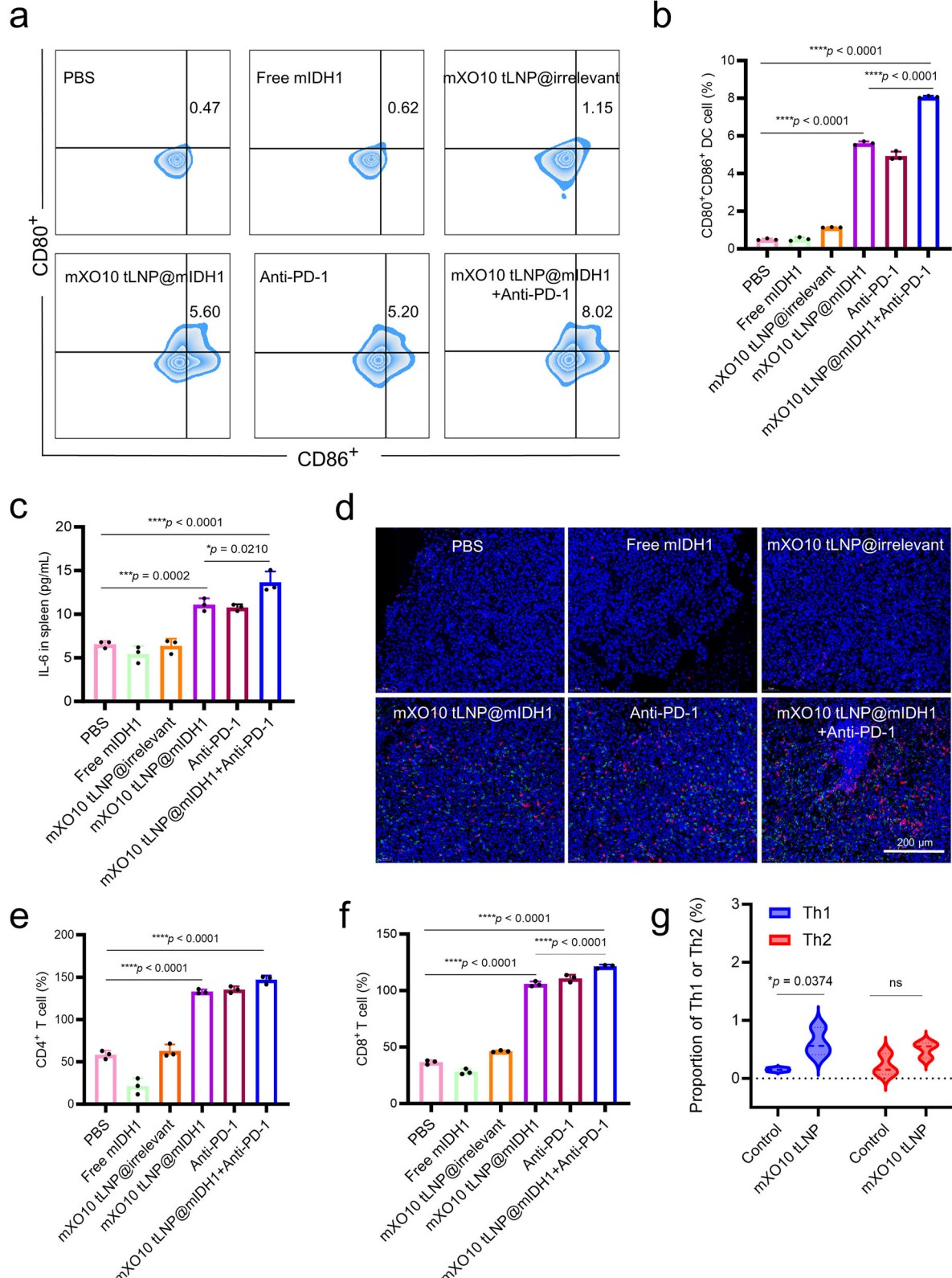

**Fig. 6 | Exploration of the mechanism of mRNA vaccine combined with Anti-PD-1 therapy in treating glioma. a** Flow cytometry analysis of the activation of splenic DC cells after different treatments. **b** Semi-quantitative analysis of DC cells activation from (**a**). **c** Measurement of cytokine IL-6 in mouse spleens after different treatments using ELISA. **d** Analysis of CD4+ T cell (red) and CD8+ T cell (green) infiltration in glioma on day 14 after different treatments. **e** Semi-quantitative analysis of CD4+ T cell infiltration in glioma from (**d**). **f** Semi-quantitative analysis of CD8+ T cell infiltration in glioma from (**d**). (n = 3 independent experiments, data were presented as mean ± SEM. For the analysis in (**b**, **c**, **e**, **f**), a one-way ANOVA followed by Tukey post hoc test was used, 95% confidence interval.) **g** The induction of specific cellular responses by mXO10 tLNP@mIDH1. Flow cytometry analysis of Th1 and Th2 cell immune subtypes. (n = 3 independent experiments, data are presented as mean ± SEM, and analyzed using two-way ANOVA followed by Tukey post hoc test was used, 95% confidence interval, *p < 0.05, **p < 0.01, ***p < 0.001, ****p < 0.0001, n.s. not significant.) Source data and exact p values are provided in the Source data file.

amounts of pro-inflammatory cytokines, thereby activating the anti-tumor immune response. CD8$^+$ T cells (cytotoxic T lymphocytes) can directly kill targeted tumor cells, while CD4$^+$ T cells (helper T cells) play an important role in adaptive immune regulation and are key in tumor immunotherapy. We studied the infiltration of CD8$^+$ T cells and CD4$^+$ T cells in the glioma. On the 14th day after treatment, glioma tissues were collected from the mice, and the infiltration of CD8$^+$ T cells and CD4$^+$ T cells in the glioma was analyzed using immunohistochemistry. The result showed a significant increase in the infiltration of CD8$^+$ T cells (green) and CD4$^+$ T cells (red) in the tumor area in the mXO10 tLNP@mIDH1 vaccine group and the Anti-PD-1 group compared to the control group (Fig. 6d–f). The infiltration of CD8$^+$ T cells in the mXO10 tLNP@mIDH1 mRNA vaccine group was approximately three times higher than that of the control group. Notably, the highest levels of CD8$^+$ T cell and CD4$^+$ T cell infiltration were observed in the group receiving the combination of the mXO10 tLNP@mIDH1 vaccine and Anti-PD-1, with CD8$^+$ T cell infiltration being approximately 3.5 times higher than in the control group. These results indicated that the mXO10 tLNP@mIDH1 mRNA vaccine and its combination with Anti-PD-1 effectively promoted the infiltration of CD8$^+$ T cells and CD4$^+$ T cells in the glioma tissue. Flow cytometry was used to analyze CD3$^+$CD8$^+$IFN-γ$^+$ T cells in the spleen on the 14th day after treatment. The result indicated that the number of CD3$^+$CD8$^+$IFN-γ$^+$ T cells and the levels of IFN-γ were significantly higher in the mXO10 tLNP@mIDH1 mRNA vaccine group and the Anti-PD-1 treatment group compared to the control group, with the number of CD3$^+$CD8$^+$IFN-γ$^+$ T cells in the mXO10 tLNP@mIDH1 mRNA vaccine group being approximately 3 times that of the control group (Figs. S28 and S29). The combination of mXO10 tLNP@mIDH1 mRNA vaccine and Anti-PD-1 had an even greater effect, increasing the number of CD3$^+$CD8$^+$IFN-γ$^+$ T cells in glioma tissue by approximately 4.4 times compared to the control group. We evaluated the induction of specific cellular responses by mXO10 tLNP@mIDH1 in a mouse glioma model and subsequently detected Th1 and Th2 cells using flow cytometry. The results showed that, compared with the control group, the proportion of Th1 cells in the mXO10 tLNP@mIDH1 group increased significantly, whereas the proportion of Th2 cells did not differ significantly. These findings suggest that mXO10 tLNP@mIDH1 immunization induced a Th1-biased cellular immune response (Figs. 6g and S30). In conclusion, the mXO10 tLNP@mIDH1 mRNA vaccine effectively stimulated DC cells activation and enhanced T cells infiltration in glioma, thereby activating the anti-glioma immune response. This effect was particularly pronounced when the mRNA vaccine was combined with Anti-PD-1, inducing a powerful anti-glioma immune response.

In addition, we evaluated the biological safety of the mXO10 tLNP@mIDH1 mRNA vaccine in vivo. As shown in Fig. S31, no abnormal histologic changes were observed in the heart, liver, spleen, or kidney in all groups, suggesting the good biocompatibility of mXO10 tLNP@mIDH1 vaccine and mXO10 tLNP@mIDH1 vaccine plus Anti-PD-1. Furthermore, serum biochemistry parameters, including albumin (ALB), alanine aminotransferase (ALT), creatinine (CREA), and glucose (GLU), were detected, and the results showed that the blood bio-chemistry parameters in the control group, the mXO10 tLNP@mIDH1 vaccine group, and the mXO10 tLNP@mIDH1 vaccine plus Anti-PD-1 group were all within the safe range, indicating the good biosafety of this mRNA vaccine and its potential for future clinical application (Fig. S32).

## Discussion

Scientists often improve the efficacy of small molecules by adjusting their molecular conformation. Here, we provide evidence that similar efforts can improve the delivery of mRNA using lipid nanoparticles (LNP). These data included the efficiency of LNP delivery within cells, their biodistribution, organ-selective and cell-targeted mRNA expression, as well as mRNA-mediated protein production, which led us to

several conclusions. First, cells exposed to meta-tLNP reacted differently compared to cells exposed to ortho-/para-tLNP, with meta-tLNP showing higher mRNA delivery efficiency than ortho-/para-tLNP. Given the consistent composition of tLNP, except for the positional isomers of ionizable lipidoids, these data suggested that cells could detect and respond to differences in the positional isomers of tLNP. It was noteworthy that the meta-/ortho-/para-tLNP exhibited similar biophysical properties, indicating that significant changes in size did not drive this effect. Secondly, these data indicated that the mechanism was biological. Specifically, the main difference between meta-/ortho-/para-tLNP lay in cellular uptake, with meta-tLNP being more likely to be internalized through endocytosis, ortho- and para-tLNP primarily through clathrin-dependent endocytosis. In vivo data supported these observations. The third observation was that there was no significant difference in biodistribution across the mouse body between meta-/ortho/para-tLNP and commercial LNP. However, while commercial LNP selectively expressed mRNA in organs such as the liver, meta-/ortho-/para-tLNP selectively expressed mRNA only in the spleen. This data reinforced the concept that while biodistribution was necessary, it was not sufficient to achieve functional mRNA delivery. The key lay in where the LNP cargo went and the specific biological effects it had on the cells.

It must have been acknowledged that this study had limitations. First, although the data on tLNP-mediated mRNA delivery were obtained from mice, the relationship between tLNP-mediated mRNA delivery and different species required further investigation. In vivo studies in non-human primates or human clinical trials might have yielded different results. Secondly, it is unclear whether similar positional isomer design rules apply to polymeric nanoparticles, extracellular vesicles, PEG10, and other non-viral delivery systems. Finally, mXE14 tLNP exhibited excellent mRNA delivery efficiency at the cellular level. However, mXE14 tLNP had low efficiency in delivering mRNA in vivo. There was a discrepancy between in vitro and in vivo mRNA delivery efficiency of tLNP. Despite these limitations, these data provided evidence for an important concept: when designing tLNP, the spatial isomeric conformation of tLNP should be considered.

Currently, most developed LNP target the liver after systemic administration. However, to fully realize the potential of mRNA therapies, SSOCT strategy is needed to direct LNP to extrahepatic organs and specific cells within these organs. Here, we demonstrated that by simply altering the chemical structure of the ionizable lipidoids and applying the SSOCT strategy, tLNP could be precisely tuned to shift organ-selective mRNA expression from the liver to the spleen, further achieving targeted mRNA expression in DC cells within the spleen. This was advantageous for developing tumor mRNA vaccines and reducing liver toxicity. This study demonstrated a strong structure-activity relationship between the chemical structure of ionizable lipidoids and organ-selective and cell-targeted mRNA delivery. We believed this research provided insights into designing organ-selective/cell-targeted LNP through simple chemical modifications, thereby advancing mRNA therapies.

The efficacy and side effects of mRNA vaccines were largely determined by the mRNA delivery system. LNP were among the most attractive mRNA delivery systems. Traditional LNP consisted of four components: ionizable lipidoids, phospholipids, cholesterol, and PEG-lipids. However, PEG-lipids could induce allergic reactions in individuals with pre-existing PEG-specific antibodies, posing life-threatening risks. Thus, the traditional four-component LNP delivery system entailed certain risks. We introduced phenyl rings and hydroxyl functional groups into ionizable lipidoid molecules to regulate LNP stability and biocompatibility, thereby successfully constructing a three-component LNP (tLNP) without PEG-lipids, thus avoiding severe allergic reactions associated with LNP.

This study showed a strong correlation between the positional isomers of ionizable lipidoids and the efficiency of mRNA delivery by

LNP, and demonstrated that the SSOCT strategy enabled tLNP to achieve targeted mRNA expression in specific cells within target organs. We synthesized meta-/ortho-/para-ionizable lipidoids using m/o/p-xylylenediamine and fatty carbon chains, and we prepared three-component LNP (tLNP) using meta-/ortho-/para-ionizable lipidoids, phospholipids, and cholesterol for mRNA delivery. We found that meta-tLNP exhibited higher mRNA delivery efficiency both in vitro and in vivo compared to ortho-/para-tLNP. Furthermore, these tLNP exhibited splenic dendritic cells-targeted mRNA expression via SSOCT and demonstrated good biocompatibility. This suggested that the structural and spatial isomers of ionizable lipidoids had a profound impact on the mRNA delivery efficiency, organ-selective mRNA expression, and cell-targeted mRNA expression of tLNP. Subsequently, we used mXO10 tLNP to develop a glioma mRNA vaccine (mXO10 tLNP@mIDH1), which demonstrated excellent anti-glioma effects. Notably, when combined with Anti-PD-1 treatment, it exhibited even stronger anti-glioma effects and was able to eliminate the glioma in mice. We believed that this study provided valuable insights into achieving extrahepatic organ-selective mRNA expression, active targeting of specific cells within the targeted organ, efficient mRNA delivery, and high biocompatibility through simple chemical design, optimization of tLNP components, and the SSOCT strategy, which would help develop tumor mRNA vaccines and other mRNA therapeutics for various diseases.

## Methods

### Cells, viruses, and plasmids

HEK293T cells, DC2.4 cells, and GL261 cells were obtained from the American Type Culture Collection (ATCC). All eukaryotic cell lines used were authenticated by short tandem repeat (STR) analysis and routinely screened for mycoplasma contamination via loop-mediated isothermal amplification (LAMP) to ensure the absence of mycoplasma. All cells were cultured in Dulbecco's modified Eagle's medium (DMEM) supplemented with GlutaMAX, 10% fetal bovine serum (FBS), and 1% penicillin-streptomycin. The lentiviral plasmids were constructed and produced by Getein Biotech, Inc.

### Animals

Male BALB/c mice (6–8 weeks old, 20 g) and male C57 mice (6–8 weeks old, 20 g) were bred in a temperature- and humidity-controlled SPF condition (20–25 °C and 45–55% humidity) under 12/12 h light/dark cycles with free access to food and water. The maximum tumor volume/load approved by the Sun Yat-sen University Ethics Committee is that tumor growth must not exceed 10% of the animal's original body weight and that the average tumor diameter in mice must not exceed 20 mm; we confirm that these limits were not exceeded during the experiment. All animal husbandry and experimental procedures were approved by the Animal Ethics Committee of the Laboratory Animal Research Center of Sun Yat-sen University.

### Synthesis of ionizable lipidoids

Meta/ortho/para-xylylenediamine (Sigma-Aldrich) was individually reacted with 1,2-epoxydecane, 1,2-epoxydodecane, 1,2-epoxytetradecane, 1,2-epoxyhexadecane, or 1,2-epoxyoctadecane (Sigma–Aldrich) at a molar ratio of 1:2.2 in 95% ethanol at 85 °C for 48 h. This series of reactions resulted in the crude products mXO8, mXO10, mXO12, mXO14, mXO16, oXO8, oXO10, oXO12, oXO14, oXO16, pXO8, pXO10, pXO12, pXO14, and pXO16, respectively. Similarly, meta/ortho/para-xylylenediamine was reacted with decyl acrylate, dodecyl acrylate, tetradecyl acrylate, hexadecyl acrylate, octadecyl acrylate, or docosyl acrylate at a molar ratio of 1:2.2 in isopropanol at 85 °C for 48 h. This process yielded the crude products mXE10, mXE12, mXE14, mXE16, mXE18, oXE10, oXE12, oXE14, oXE16, oXE18, pXE10, pXE12, pXE14, pXE16, and pXE18, respectively. The m/o/pXO10 and m/o/pXE14 compounds obtained from the above reactions were purified using silica gel column chromatography. Subsequently, the purified compounds were subjected to structural verification through Nuclear Magnetic Resonance ($^1$H-NMR) spectroscopy.

### The preparation of tLNP and screening of mRNA-loaded tLNP

Preparation of m/o/pXO(n + 1) tLNP@mLuc (n = 7, 9, 11, 13, 15): The m/o/pXO(n + 1) ionizable lipidoids, DOPE (Avanti Polar Lipids), and cholesterol (Sigma–Aldrich) were mixed in ethanol at a molar ratio of 9:6:4. Luciferase mRNA (mLuc) was dissolved in PBS buffer. The ethanol phase and aqueous phase were mixed at a ratio of 1:9 to obtain m/o/pXO(n + 1) tLNP@mLuc with varying N:P ratios. Similarly, the preparation of m/o/pXE(n + 3) tLNP@mLuc (n = 7, 9, 11, 13, 15) was conducted, but the mLuc was dissolved in a pH 4 citrate-sodium citrate buffer instead of PBS. This variation in pH conditions affected the interaction of tLNP with mRNA. Preparation of mXO10 tLNP@OligoT: mXO10, DOPE, and cholesterol were dissolved in ethanol at a molar ratio of 9:6:4. OligoT (a single-stranded nucleic acid sequence of 50 Ts) was dissolved in PBS buffer. The ethanol and aqueous phases were mixed at a 1:9 ratio to obtain mXO10 tLNP@OligoT. Preparation of mXO10 tLNP@mIDH1 mRNA vaccine: mXO10, DOPE, and cholesterol were dissolved in ethanol at a molar ratio of 9:6:4, and the IDH1(R132H) mRNA was prepared in PBS. The two phases were mixed at a 1:9 ratio, specifically targeting an N:P ratio of 3, to obtain the mXO10 tLNP@mIDH1 mRNA vaccine. Preparation of mXO10/mXE14 tLNP@Cy5-labeled mRNA: First, Cy5-labeled OligoT-18 (a single-stranded sequence of 18 Ts) was hybridized with mLuc by incubating them together at 37 °C for 15 min, allowing the OligoT-18 to bind to the poly-A tail of the mRNA. The tLNP preparation was then carried out following either the (1) or (2) protocol, incorporating the Cy5-labeled mLuc. Screening of tLNP in HEK-293 Cells: HEK-293 cells were seeded in a white-bottomed 96-well cell culture plate at a density of 20,000 cells per well. After 24 h of incubation, the prepared tLNP was added to the wells and co-incubated for another 24 h. Then, 100 µL of One-umi™ firefly luciferase reporter gene assay reagent (Beyotime Biotechnology) was added to each well, and the plate was gently mixed on a shaker for 10 min. The bioluminescence intensity of the cells was measured using a multimode microplate reader to evaluate the efficiency of tLNP delivery of mRNA.

### Determination of the pKa value of tLNP

A series of buffer systems with pH values ranging from 3 to 12 were prepared, and a 300 µM solution of TNS (2-(p-toluidino) naphthalene-6-sulfonic acid) (Sigma-Aldrich) was freshly prepared in DMSO (Sigma–Aldrich). Additionally, mXO10 tLNP@mLuc and mXE14 tLNP@mLuc were prepared. Different pH buffer solutions were dispensed into a black-bottomed 96-well plate (90 µL per well), followed by the addition of 10 µL of tLNP solution to each well, and the mixtures were thoroughly mixed. Subsequently, 2 µL of TNS was added to each well. The fluorescence intensity of each well was measured at an excitation wavelength of 325 nm and an emission wavelength of 435 nm using a microplate reader. The obtained fluorescence values were then plotted against the pH of the buffer solutions to generate an S-shaped curve. The logarithm of the pH value at the inflection points of this curve corresponded to the pKa value of tLNP.

### The mRNA encapsulation efficiency of mXO10 tLNP and mXE14 tLNP was evaluated

An agarose gel was prepared with the addition of GEL-RED nucleic acid stain. The samples were mixed with loading buffer, and the following were loaded into separate wells: 200 ng of free luciferase mRNA (mLuc), mXE14 tLNP@mLuc + 200 ng/µL RNase, mXO10 tLNP@mLu + 200 ng/µL RNase, mXE14 tLNP@mLuc, and mXO10 tLNP@mLuc (tLNP containing 200 ng of mLuc). Electrophoresis was performed at a constant voltage of 110 V until the bromophenol blue

dye migrated to approximately 1 cm from the bottom of the gel. A gel documentation system was used to image the gel.

## Cellular uptake and lysosomal escape
DC2.4 cells and GL261 cells (8000 cells per well) were seeded into an 8-well chamber (Sigma-Aldrich). After 24 h of incubation, mXO10 tLNP@Cy5-labeled mRNA, mXE14 tLNP@Cy5-labeled mRNA, and free mRNA (each containing 400 ng of mRNA) were added to the respective wells and incubated for 1.5 h. The media were aspirated, and the cells were washed with PBS. The cells were then stained with Hoechst 33342 for 15 min, followed by aspiration and three washes with PBS. Next, 100 μL per well of Lysotracker Green (Thermo Fisher Scientific) staining solution (diluted 10,000-fold) was added and incubated for 30 min, followed by aspiration and further PBS washes. The prepared samples were imaged using a confocal laser scanning microscope.

## In vivo mRNA delivery and distribution of mXO10 tLNP and mXE14 tLNP were evaluated
BALB/c mice were injected with mXO10 tLNP@mLuc, mXE14 tLNP@mLuc, and free mLuc (at a dose of 0.5 mg/kg) via the tail vein. Bioluminescence imaging was performed at 1, 3, 6, 12, 24 h post-injection to assess the efficiency of mRNA delivery by tLNP. Additionally, BALB/c mice were injected with mXO10 tLNP@Cy5-labeled mRNA, mXE14 tLNP@Cy5-labeled mRNA, and free Cy5-labeled mRNA (at a dose of 0.1 mg/kg) via the tail vein. Fluorescence imaging was conducted using a small animal live imaging instrument at 1, 3, 6, 12, 24 h post-administration to examine the biodistribution of the tLNP.

## Construction of GL261-*IDH1 R132H* cell line
(a) 293 T cell seeding: Seed 293T cells in a 6-well plate and culture overnight until the confluence reaches 70–90%; (b) Transfection mixture preparation: (1) Solution A: Mix the packaging plasmid, envelope plasmid, and transfer plasmid (containing the target gene) in a ratio of 0.8 μg: 0.4 μg: 1.2 μg. Add them to 100 μL of opti-MEM (serum-free and antibiotic-free), mix gently 20 times, and let it stand at room temperature for 5 min; (2) Solution B: Add 7.2 μg of PEI reagent to 100 μL of opti-MEM, mix gently 20 times, and let it stand at room temperature for 5 min; (3) Mixing and transfection: Add solution B dropwise to solution A, mixing gently after each addition. Let it stand at room temperature for 30 min. Discard the old culture medium from the 6-well plate, rinse with PBS, and add 2 mL of fresh complete culture medium. Gently add 200 μL of the transfection mixture to the cells, and gently shake the culture plate to evenly distribute the transfection mixture. Continue culturing the cells; (4) Virus collection: After 24 h, observe the expression of fluorescent protein. Collect the cell supernatant at 48 h and 72 h. Centrifuge at 4 °C and $845 \times g$ for 10 min to remove cell debris, and filter through a 0.22 μm filter membrane; (5) Target cell seeding and infection: Seed GL261 cells in a culture plate. When the confluence reaches 30–40%, infect the cells with the virus, and add polybrene to a final concentration of 5 μg/mL. After 24 h, replace with fresh culture medium containing 10% FBS; (6) Stable cell line screening: After observing the expression of fluorescent protein, add puromycin to a final concentration of 10 μg/mL for selection. Repeated selection yields the GL261-IDH1 stable cell line.

## Establishment of orthotopic glioma model
Male C57 mice were anesthetized via intraperitoneal injection of 1% sodium pentobarbital at a dose of 50 mg/kg and then fixed onto the base of a brain stereotaxic apparatus. The head epidermis was disinfected 2–3 times with iodine and alcohol, followed by a 1.5 cm skin incision. After gently wiping the skull surface with hydrogen peroxide, the positioning needle was placed at the bregma point, designated as X = 0, Y = 0, Z = 0. At coordinates X = −1.6, Y = −4.6, a 1.5 mL syringe was used to puncture the dura mater. The positioning needle aspirated approximately 500,000 GL261 cells and was moved to X = −1.6,

Y = −4.6, Z = −3. The GL261 cells, labeled with firefly luciferase, were implanted intracranially at a rate of 5 μL every 15 s. After the injection, the positioning needle was left in place for 3 min before being withdrawn at a speed of 0.01 mm/s. The skull surface was disinfected, and the skin was sutured.

## The mXO10 tLNP@mIDH1 mRNA vaccine and its combination with PD-1 antibody (Anti-PD-1) were tested for treating glioma
On the 8th day after modeling, bioluminescence imaging was performed using a small animal live imaging system. C57 mice with bioluminescence intensity greater than $5.0 \times 10^7$ and less than $5.0 \times 10^8$ were selected and randomly divided into the following groups: PBS group, Free mIDH1 group, mXO10 tLNP@irrelevant group, mXO10 tLNP@mIDH1 mRNA vaccine group, Anti-PD-1 group, and mXO10 tLNP@mIDH1+Anti-PD-1 combination group. The mXO10 tLNP@mIDH1 mRNA vaccine was administered intravenously at a dose of 0.05 mg/kg on days 8, 12, and 16. Anti-PD-1 was administered intraperitoneally on days 7, 11, 14, and 18. Bioluminescence imaging was used to monitor glioma cells growth, mouse survival time, and other indicators to evaluate the efficacy of glioma mRNA vaccine therapy and its combination with Anti-PD-1 therapy.

## In vivo delivery of mXO10 tLNP to different immune cells in the spleen was studied
mXO10 tLNP encapsulated with tdTomato mRNA was prepared using mXO10 as the ionizable lipid at an N:P ratio of 3. C57 mice were intravenously injected with the prepared mXO10 tLNP@ mtdTomato at a dose of 0.2 mg/kg. After 24 h, the spleen was harvested, minced, and passed through a 70 μm filter to create a cell suspension. The cell suspension was then lysed with red blood cell lysis buffer for 10 min on ice, followed by washing with PBS. Different types of immune cells were labeled using specific antibodies: DC cells were labeled with CD11c antibody, macrophages were labeled with CD11b and F4/80 antibodies, NK cells were labeled with NK1.1 antibody, and T cells were labeled with CD3 antibody. The staining process was performed on ice for 15 min, followed by centrifugation at $135 \times g$ for 5 min to remove the supernatant. The cells were then washed once with PBS and filtered through a 300-mesh sieve before being analyzed on a flow cytometer. The flow cytometer was used to detect the expression of tdTomato mRNA in different types of immune cells isolated from the spleen. This analysis allowed us to investigate the delivery and expression of the mRNA by tLNP in specific immune cell populations.

## Flow cytometric analysis of tdtomato expression in different cell types in the spleen of Ai9 mice
The mXO10 tLNP@mCre was prepared using the aforementioned method with mXO10 as the ionizable lipidoids and encapsulating Cre mRNA (mCre) (N:P ratio = 3). Ai9 mice were intravenously injected with mXO10 tLNP@mCre at doses of 0.5 mg/kg and 0.2 mg/kg, respectively. After 48 h, the spleens were harvested, homogenized, and filtered through a 70-μm cell strainer to prepare a single-cell suspension. The cell suspension was treated with red blood cell lysis buffer for 3 min on ice. Subsequently, Fc receptor blocking solution was added and incubated on ice in the dark for 10 min. Different types of immune cells were labeled as follows: DC cells were labeled with CD11c antibody, macrophages were labeled with CD11b and F4/80 antibodies, NK cells were labeled with NK1.1 antibody, B cells were labeled with CD19 antibody, and T cells were labeled with CD3 antibody. During the staining process, the cells were kept on ice. After 15 min, the supernatant was removed by centrifugation at $135 \times g$ for 5 min. The cells were washed once with PBS and then filtered through a 300-mesh sieve. The expression of tdTomato in different types of immune cells in the spleen was detected by flow cytometry.

## Analysis of DC cells activation

On the 21st day after modeling, brain and spleen tissues were harvested from mice in different treatment groups. The tissues were minced and passed through a 70 μm filter to create cell suspensions. For spleen tissue, the cell suspension was treated with red blood cell lysis buffer on ice for 10 min before washing with PBS. The cells were then stained simultaneously with CD80 and CD86 antibodies on ice for 15 min. After staining, the cells were centrifuged at $135 \times g$ for 5 min to remove the supernatant. The cells were washed once with PBS and filtered through a 300-mesh sieve before being analyzed on a flow cytometer. The number of CD80$^+$CD86$^+$ cells in the spleen and brain tissue samples from different treatment groups of C57 mice was quantified by flow cytometry.

## Analysis of cytokine levels by ELISA

On the 21st day after modeling, spleen tissues were harvested from C57 mice in different treatment groups. The tissues were homogenized using a high-speed tissue grinder in PBS, with a tissue weight to volume ratio of 1:9, and protease inhibitors were added. This process resulted in tissue homogenates. ELISA kits were used to quantify the levels of tumor necrosis factor-α (TNF-α), Interleukin-6 (IL-6), Interleukin-12 (IL-12), and Interferon-γ (IFN-γ) in the spleen tissue homogenates from the various groups.

## Immunohistochemical analysis of T cells activation in glioma

C57 mice administered with different treatments were sacrificed on the 21st day after modeling, and their brain tissues were harvested. The tissues were fixed in 4% paraformaldehyde for 24 h. Subsequently, the tissue samples underwent gradual dehydration using varying concentrations of ethanol (70%, 80%, 95%, and 100%) before being cleared in xylene. The dehydrated tissues were then infiltrated with molten paraffin at 60–65 °C, embedded in paraffin blocks, and sectioned using a rotary microtome. The sections were mounted on glass slides, dewaxed with a clearing agent, and rehydrated. The paraffin sections were subjected to antigen retrieval by immersing them in 0.01 M citrate buffer (pH 6.0) and heating in a microwave oven at high power until boiling, followed by 15 min at medium power. The sections were then blocked with TBS containing 10% normal serum and 1% BSA for 2 h at room temperature. Overnight incubation with specific antibodies against T cell markers CD4 and CD8 was performed at 4 °C. The slides were washed twice in TBS containing 0.025% Triton X-100 for 5 min, followed by incubation with a fluorescently labeled secondary antibody diluted in TBS with 1% BSA for 1 h. After secondary antibody incubation, the slides were washed three times with TBS for 5 min. Nuclear staining was performed using DAPI for 15 min, followed by three washes with PBS. Finally, the slides were mounted with a coverslip using a mounting medium and observed under a fluorescence microscope for imaging.

## Analysis of activated cytotoxic T lymphocytes (CTLs) secreting IFN-γ

On the 21st day after modeling, spleen tissues were harvested from C57 mice in different treatment groups. The tissues were minced and passed through a 70 μm filter to create a cell suspension. Erythrocytes were lysed, and the lysing solution was removed. The cells were resuspended in PBS at a tissue weight to volume ratio of 1:19. Stimulatory agents and protein transport inhibitors, including phorbol 12-myristate 13-acetate (PMA), ionomycin, monensin, and Brefeldin A (BFA), were added to the cell suspension. The cells were incubated at 37 °C for 5 h to stimulate intracellular cytokine production. After incubation, the cells were centrifuged at $135 \times g$ for 5 min to remove the supernatant. Fc receptor (FcR) block (1 μL per tube) was added to the cell pellet and incubated at room temperature for 10 min to block non-specific binding. CD3 and CD8 antibodies were then added to the samples and incubated on ice for 15 min to label the specific T cell

subsets. The cells were centrifuged at $135 \times g$ for 5 min, and the supernatant was discarded. To prepare the cells for intracellular staining, 300 μL of fixation buffer was added, and the cells were incubated in the dark at room temperature for 20 min. After fixation, the cells were washed and permeabilized using 100 μL of a 1:10 dilution of intracellular staining perm wash buffer. Next, 1 μL of IFN-γ antibody was added, and the cells were incubated on ice in the dark for 50 min to stain for intracellular IFN-γ. Following antibody incubation, the cells were centrifuged to remove the supernatant and washed once with PBS. The cells were resuspended in 300 μL of PBS and analyzed using a flow cytometer to quantify the number of CD3$^+$CD8$^+$IFN-γ$^+$ cells within the spleen tissue.

## Analysis of hepatic and renal toxicity of tLNP in mice

On the 21st day post-modeling, blood samples were collected from the orbital sinus of C57 mice in different experimental groups. The whole blood samples were allowed to stand at 4 °C for 8 h before being centrifuged at $845 \times g$ and 4 °C to separate the serum. The obtained serum was then stored at −80 °C to prevent repeated freeze-thaw cycles. Parameters were set on a fully automated biochemistry analyzer, and the samples were loaded for automatic measurement of alanine aminotransferase (ALT), aspartate aminotransferase (AST), albumin (ALB), urea (UREA), creatinine (CREA), and uric acid (UA) levels in the serum.

## Hematoxylin and eosin staining of major organs in mice

On the 21st day post-modeling, mice were euthanized, and their hearts, livers, spleens, lungs, and kidneys were harvested. The organs were promptly fixed in 4% paraformaldehyde for 24 h. After fixation, the tissues were processed to create paraffin blocks. Using a microtome, sagittal sections of 5–8 μm thickness were cut from each organ block. The paraffin was removed from the sections with xylene, and the sections were then passed through a series of alcohol solutions from high to low concentrations, ending in distilled water. The deparaffinized and rehydrated tissue sections were immersed in hematoxylin solution for 5 min to stain the nuclei. Following this, the slides were briefly differentiated in acid alcohol and ammonia water for approximately 10 s each to enhance the nuclear staining contrast. The sections were rinsed under running tap water for 1 h and then briefly dipped in distilled water. To dehydrate the tissue, the slides were placed in 70% and 90% alcohol solutions for 10 min each. Finally, the sections were stained with eosin solution for 3 min to impart a pinkish-red color to the cytoplasm and other non-nucleated structures.

## Statistical analysis

The analysis tools used included one-way ANOVA (Analysis of Variance) and two-way ANOVA (analysis of variance). Results are presented as mean ± SEM (standard error of the mean) to provide a comprehensive understanding of data distribution and variability. Statistical significance is indicated as follows: *$p < 0.05$, **$p < 0.01$, ***$p < 0.001$, ****$p < 0.0001$.

## Ethical approval

This study complies with all relevant ethical regulations at Sun Yat-sen University and was approved by the Institutional Review Board under ethical permit AEDGJ202301. All experimental animals were purchased from Zhuhai BesTest Biotechnology Co., Ltd.

## Reporting summary

Further information on research design is available in the Nature Portfolio Reporting Summary linked to this article.

# Data availability

The data supporting the findings of this study are included in the main text, Supplementary Information file, and Source Data. The original

images of the NMR spectra are accessible in the Source data. Source data file is provided with this work. Source data are provided with this paper.

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

## Acknowledgements

This work was supported by Shenzhen Science and Technology Program (Shenzhen Key Laboratory of Neural Cell Reprogramming and Drug Research, ZDSYS20230626091202006) to G.J.D., Natural Science Foundation of Guangdong Province (2024A1515012496) to G.J.D., Shenzhen Science and Technology Program (KQTD20190929173853397) to W.B.D., Guangdong Basic and Applied Basic Research Foundation (2021A1515220108) to G.J.D., Shenzhen Outbound Postdoctoral Research Funding Project (szbo202208) to G.J.D., and National Natural Science Foundation of China (32000982) to G.J.D.

## Author contributions

L.S. and Y.Y.L.: Data collection, experimentation, data analysis, writing: original draft; S.Y.H., J.P., C.Y.L., and Z.Y.H.: Data analysis, review and editing; G.J.D. and W.B.D.: Data analysis, Writing: review and editing, supervision, acquiring of funding. All authors contributed intellectually and approved the final manuscript for publication.

## Competing interests

The authors declare no competing interests.
