## [Transparent Peer Review file · Nature Communications]

mRNA Vaccine developed for Sequential Selective Organ-to-Cell Targeting of Glioma

Corresponding Author: Dr Guanjun Deng

Version 0:

Reviewer comments:

Reviewer #1

(Remarks to the Author)

This article employs innovative methods to design LNPs targeting the spleen and dendritic cells (DCs) for mRNA delivery and successfully develops a glioma mRNA tumor vaccine that offers protection to mice. However, the article contains significant logical flaws and several basic errors. If these issues are not addressed, it is not recommended to accept this article for publication.

Major comments :

1. There are already many studies on spleen-targeted LNPs, such as LNP surface modification with antibodies, specific ligand targeting, or using SORT LNP strategies, and achieving spleen targeting by incorporating anionic lipids. In this study, mxO10 has a zeta potential of -38.38mV, and mxE14 is 2.8mV, both of which can target the spleen. The paper should explain what mechanism the strategy in this work uses to achieve spleen targeting.
2. The layout of the paper is not logical. Figure 1 talks about the screening of ionizable lipid libraries and lysosomal escape in DC cells, while Figure 3 discusses LNP preparation and characterization and repeats testing of lysosomal escape in DC cells. Please rearrange the layout for clarity.
3. mxO10 and mxE14 are considered the most efficient delivery lipids only based on Figure 1. In Figure 1b, many lipids show similar delivery efficiency to mxO10, so why are they not effective?
4. In Figures 2 and 3, data for mxO10 and mxE14 are sometimes compared, and sometimes mxO10 is discussed separately. Please organize the figures and their logic. If mxO10 is the best lipid, then do not repeat tests or comparisons.
5. The DLS results in Figure 3 show that the particle size is around 300 nm. Please aim for a size of around 100 nm and observe the in vivo distribution. Additionally, add data on the PDI, thermal stability, and encapsulation efficiency of the LNPs. In Figure 3e, the gel electrophoresis should include an RNA marker, and please refer to <https://doi.org/10.1038/s41541-022-00549-y>.
6. The mechanistic exploration in Figure 5 is superficial and does not test specific cellular immune responses. It only looks at general T-cell responses. Please add tests for specific cellular immune responses.
7. The paper emphasizes the safety of spleen targeting because there is no accumulation in the liver and kidneys. However, does spleen accumulation increase the burden on the spleen? Please provide additional evidence that the spleen's normal function is not affected beyond the current safety evaluation.

Minor comments :

1. "Notably, we demonstrated that the delivery of mRNA vaccine (mXO10 tLNP@mIDH1) using meta-tLNP was successful in treating glioma, particularly when combined with Anti-PD-1 therapy, which further improved efficacy and even completely eradicated glioma, while reducing hepatotoxicity and minimizing PEG-lipid-induced allergic reactions." The treatment effect is only observed in mice, please correct this statement.
2. "We introduced phenyl rings and hydroxyl functional groups into ionizable lipidoids to improve LNP stability and biocompatibility." Why does adding phenyl rings and hydroxyl groups increase stability? Please explain further in the text.
3. "Traditional tumor vaccine is mainly divided into cellular vaccine, peptide vaccine, and DNA vaccine. However, these traditional tumor vaccines suffer from low safety, low protective efficiency, and long production cycles." There are already many mRNA tumor vaccines. Please include the limitations of current mRNA tumor vaccines and explain how this study can address or optimize these issues.
4. "Confocal laser scanning microscopy (CLSM) images showed no significant difference in lysosome escape among meta-, ortho- and para-tLNP (Figure 1c and S7-S9)." How was the conclusion of no significant difference made without statistical analysis? Please perform a quantitative analysis of the images (Figure 1c and S7-S9).
5. In Figure 1b, what cells were used to test the transfection efficiency? Please explain this in the figure legend and the text.

6. "In contrast, ALC-0315@mLuc showed strong protein expression levels in both the liver and kidneys." The results show expression in the liver and spleen, but not in the kidneys.
7. "Therefore, an mRNA delivery system that selectively targets DC cells in the spleen through the SSOCT strategy can not only enhance the therapeutic efficacy of mRNA vaccines but also reduce toxicity to major organs such as the liver and kidneys." Please revise this statement, as the lack of accumulation in the liver and kidneys does not necessarily imply safety.
8. (Figure S13) What method was used for detection, and what dye was used for labeling? Please clarify this in the text and figure legend.
9. In Figure 2g, why is the fluorescence of mxO10 strongest at 1 hour, and then shows a trend of decreasing, followed by an increase, and then a decrease again? Please check the image and create a fluorescence intensity-time curve. The figure legend is missing.
10. Data in Figure 3f and Figure 1 are repeated; 3g and supplementary Figure S17b are also repeated. The time points on the figure need to be labeled, and the molecular weight (kDa) should be indicated in the figure.
11. Please perform quantitative analysis on Figure 4f.
12. The y-axis in Figure 5g should represent T-cell statistical analysis, not DC cells.

Reviewer #2

(Remarks to the Author)

This study developed a three-component mRNA delivery system (tLNP) with selective expression in dendritic cells, enhancing tumor mRNA vaccine development and reducing toxicity. The study's main conclusion is that positional isomerism of ionizable lipidoids in multi-component lipid nanoparticles significantly impacts mRNA delivery efficiency and stability. Specifically, the meta-tLNP version showed significantly higher mRNA delivery efficiency than the ortho- and para-tLNP. Interestingly, the main difference between meta-/ortho-/para-tLNP may be in the cellular uptake mechanism, with meta-tLNP being more likely to be internalized through endocytosis, ortho- and para-tLNP primarily through clathrin-dependent endocytosis. When compared with other mRNA-delivering systems, the mxO10 tLNP delivered mRNA at 1.6 times the efficiency of lipo2000 and 1.4 times that of the commercial ALC-0315 LNP. This is an original study, and the mxO10 tLNP biodistribution data in Figure 2 look promising.

However, several major issues need to be clarified and/or addressed.

1. Multiple types of DCs reside in the spleen and secondary lymphoid organs. These DC subsets perform different immunological functions. mRNA uptake and protein expression in cDC1s, cDC2s, and pDCs should be determined (instead of using a pan DC marker such as CD11c). The mRNA uptake and expression in macrophages should also be investigated. The mRNA uptake and expression in major lymph nodes should also be determined.
2. GL261 cells are typically IDH1 wild-type (IDH1-WT). However, the authors used an mRNA vaccine encoding the IDH1 R132H mutation. This appears to be a major discrepancy which must be clarified.
3. The methods used to characterize T cell responses to the IDH1 R132H encoding mRNA vaccine must be significantly improved. The methodology for the intracellular cytokine staining assays is unclear, making it difficult to assess the immunogenicity of the vaccine.

Minor points:

- Scheme 1 is not informative.
- The Introduction section is too long and repetitive.
- The rationale for performing uptake studies in GL261 cells is not provided.

Reviewer #3

(Remarks to the Author)

This is an overall interesting manuscript with a very robust collection of data. I feel this paper could be potentially accepted and published in Nature Communications, however there are major revisions that would need to be addressed first. Although targeted mRNA delivery and vaccines remain an active area of investigation, with great room for improvement, more data would be needed to show that the current system is an improvement to current delivery vehicles. This point is not relevant for specialized journals, but for the broad audience of Nature Communications, readers would want to understand the new delivery system in context with the broader LNP and vaccine fields. I recommend to address the following points for potential reconsideration:

1. The chemical approach here (summarized in Figure 1a) is a big strength of the manuscript. The direct comparison of meta, ortho, and para is very intriguing. More could be done to further elaborate on this point, because it is a real strength of the manuscript. The compounds synthesized are highly systematic and organized with 3 core and 10 hydrophobic domains of varying length. And the lipid collection and systematic variation is much better executed than similar referenced papers. This chemistry is a strong point of the paper.
2. The proposed concept here is that meta ionizable amino lipids are superior to para and ortho ionizable amino lipids. Since this is proposed as the key advance, more mechanistic data is needed to prove that definitively. To this point, Scheme 1 introduces the concept but does not explain how or why meta would be superior.

3. Figure 1b, if meta > ortho/para, then one would expect that the heat map would show this clearly? There appears to be some hits in all three categories. The meta group has the highest overall top hit, but the general hit rate appears to be similar in total hits to ortho and para. So I found these results to be contradictory to the conclusion, and not fully supportive of the hypothesis that meta is superior. Some further investigation and/or explanation may be necessary. Again, the chemistry approach here is awesome – to examine meta/ortho/para systematically, but the current results do not seem to fully support the conclusions.

4. In the introduction and main text and results, the authors claim that endocytosis differences explain why meta lipids are better than ortho/para lipids. More depth in the mechanism is needed to support this conclusion. For example, in the Main Text description and data in Figure 3, some differences in cellular uptake are noted. What is lacking is convincing data to support the mechanism detail. The authors show differences in endocytosis through use of inhibitors, but there is no link to the LNP chemistry or physical properties. The authors should separate the “chemistry” effects (Is meta really different than ortho/para? HOW does this chemical change directly yield different delivery outcomes conclusively?) and the “physical” effects (Does meta really change the physical properties of LNPs versus ortho/para? HOW do those physical properties change and HOW does that control delivery outcomes?) and potentially “biology” effects (Can cells sense meta LNPs differently than and somehow result in differential delivery outcomes?). These questions are super interesting and important! And the authors have the chemical probes to test it (a very systematic and excellent library of lipids). The current data is unfortunately very light, superficial, and nonspecific, so the conclusions cannot be made. Some definitive and direct links need to be made through additional experiments.

5. mXO10 tLNPs are compared to ALC-0315 LNPs, with descriptions and conclusions in the Main Text summary, as well as data in some instances (e.g., Figure 2b). It is stated in the Main Text conclusions around both efficiency of delivery and tissue tropism (e.g., liver and spleen). Since luciferase bioluminescence is used as a readout, these results may be influenced by the dose. For example, at higher doses, lower expression may become detectable that would change the conclusions around tropism. This is further complicated by the efficacy differences of the LNPs tested. Therefore, it is required to perform a dose response experiment. For example, similar to Figure 2b, the authors can perform a dose response comparing mXO10 tLNPs and ALC-0315 LNPs. This will more accurately benchmark both the delivery efficiency and the tissue protein expression tropism.

6. The title of Selective Organ-to-Cell Targeting (SSOCT) may need to be changed, unless more convincing data can be generated. From my review, the results are similar to all other active and endogenous targeting LNPs. In other words, there is some organ and some cell pattern but not selective. Active (e.g. antibody conjugated) LNPs have high cell delivery (e.g. to T cells or bone marrow) but very poor organ delivery (most LNPs accumulate in liver). Endogenous (e.g. SORT) LNPs have high organ delivery (e.g. majority of injected dose accumulates in lungs or spleen) but poor cell delivery (transfects most cells in the extrahepatic organ). The authors claim that mXO10 tLNPs are different, but the current data set does not support that conclusion. mXO10 tLNPs are probably using endogenous targeting, and the claim is dendritic cell (DC) specific (e.g. Figure 2d). However, the data shown here (Figure 2d) includes very high background (the free mRNA administered mice have 7-8% TdTomato expression). These results may also be different with lower or higher injected dose. Given that only 4 cell types in the spleen were examined, and that the background signal is super high, one cannot conclude DC cell “selectivity.” The results also show that the tLNPs are not organ selective either (Figure S13). The result here is fine – I just do not think it is realistic for LNPs to be 100% cell selective, nor is there evidence of “SSOCT” in this paper. If the authors want to prove cell selectivity, then they would need to do a dose response. They would also need to quantify delivery using a more accurate method (e.g. the Ai9 or Ai14 mice transfected by Cre recombinase mRNA, or using barcoded mRNA). It would also probably be necessary to quantify delivery to more than 4 cell types (e.g. quantify by using single cell sequencing).

7. One of the amazing and striking results was described in the characterization section. “We also measured the Zeta potential of tLNP, finding that the surface potential of mXO10 tLNP@mRNA was approximately -32.38 mV, while that of mXE14 tLNP was 2.80 mV.” To me, this is a huge difference. It is furthermore amazing that changing the linker can alter surface charge this much. So I wonder if the authors can more precisely pinpoint what changes from meta to ortho/para? This could be the key to the whole study. Does meta change the LNP nanostructure? (Some images are shown in Figure 3c, but are not clear). One could use cryo-TEM and determine if meta lipids organize differently than ortho/para lipids, which could then change the surface nanostructure which could then change protein corona which could then change cellular uptake? And/or does meta change the size and/or surface charge, which could also affect protein corona or other cellular interactions? Better direct proof of what changes (meta versus ortho/para) and how that affects delivery outcomes is needed.

8. Since the tLNP concept is introduced, whereby no PEG lipid is used, more stability studies are needed. In addition to the reported 3-component LNP diameters being larger than traditional 4-component LNP diameters, one would be concerned with poor stability of PEG free LNPs. Such LNPs could aggregate in plasma, or adsorb non-specific proteins from plasma, or otherwise have reduced in vivo utility due to poor stability. Studies should be performed and added to characterize tLNP stability in depth with and without added PEG lipid.

9. The authors claim that mXO10 tLNPs are a superior platform for cancer vaccine therapy (Figure 4). To support this conclusion, the authors should directly compare mXO10 tLNPs to ALC-0315 LNPs in the glioma therapy study (Figure 4). I actually recommend to directly compare meta LNPs to ortho and para LNPs in this therapy study too. That could show superiority and uniqueness of meta lipids. But at a minimum, one would expect benchmarking to ALC-0315 LNPs if this new tLNP platform can be claimed to be better.

10. Some related literature may be cited:

- a. "Paracyclophane-based ionizable lipids for efficient mRNA delivery in vivo. *Journal of Controlled Release*, Volume 376, December 2024, Pages 395-401. This study also looked at positional isomers.
- b. "Iterative Design of Ionizable Lipids for Intramuscular mRNA Delivery" *Journal of the American Chemical Society*, Volume 145, Issue 4, Pages 2294 - 23041 February 2023. This paper examined isomeric variations in ionizable lipids, with some impact for organ-specific delivery.

11. LNPs are amorphous particles made via self-assembly. Even though the lipids used are meta/ortho/para, they are not chiral lipids per my understanding. Regardless, it is difficult to fully understand and rationalize how these isomeric and structural differences could affect downstream delivery outcomes. More thought and evidence can be applied to explain the data in this paper. Ultimately, what is the connection? Is it chemical, physical, biological? Protein corona or cell trafficking biology? How could meta/ortho/para mediate cell specific delivery? There is a lot of missing space between chemical synthesis and cell specific delivery. Explaining with data how exactly meta lipids mediate these changes would be super interesting and an amazing advance for the field!

Version 1:

Reviewer comments:

Reviewer #1

(Remarks to the Author)

While the authors have addressed the concerns raised in previous iterations, several critical issues persist that require substantive resolution.

1. The current optimized LNP preparation protocol has improved particle homogeneity, while the zeta potential of mxE14 changed dramatically from +2.8 mV to -17.95 mV. Such a significant alteration would inevitably invalidate all previously acquired experimental data. Therefore, all related experiments should be re-evaluated using the updated LNP.
2. The evaluation of antigen-specific immune cells represents a critical criterion for assessing cancer vaccine efficacy. These data should be presented in the main text rather than supplementary materials. Although the author provided data on Th1 and Th2, there are still some doubts. What are the markers selected for detecting TH1 and TH2? What is the specific operation method? What does it mean that mXo14 tLNP@mRNA 1/3 samples induced the production of IFN- γ + IL-4+ double positive? What is the logic of the flow gate? In addition, ELISPOT is also an important method for detecting specific cellular immunity.
3. For all fluorescence microscopy images and histopathological sections presented in both the main text and supplementary materials, the scale bar must be supplemented.

Reviewer #2

(Remarks to the Author)

The authors have adequately addressed some of the reviewers' concerns. The manuscript's strength lies in the innovative chemical methods used to synthesize the library of meta-, ortho-, and para-ionizable lipidoids, and the authors have excelled in this aspect. However, several issues remain regarding the immune phenotypes described in the manuscript.

One major concern is the analysis of cytotoxic T lymphocytes (Figure 5G). This analysis was conducted using non-specific stimulation with PMA and ionomycin rather than antigen-specific stimulation. As a result, no conclusions can be drawn about the immunogenicity of the vaccine based on these studies. It is essential for the authors to adopt an antigen-specific approach to measure mutant IDH-specific CD8+ T cells. Typically, vaccine-elicited CD8+ T cells should recognize an epitope corresponding to the IDH1 R132H mutation. Therefore, the ICS assay should utilize peptides containing this mutated epitope instead of relying on PMA and ionomycin. The immunogenicity of the new mRNA-LNP vaccines cannot be accurately assessed without measuring antigen-specific CD8+ T cell responses, ideally using commercial LNPs as a benchmark.

Additionally, the significance of the finding stating, "These results indicated that mXO10 tLNP significantly increased mRNA expression specifically in cDC2s," is unclear in the context of anti-tumor immunity. cDC1s are critical for mediating CD8+ T cell anti-tumor responses and play pivotal roles in priming CD4 T cell responses. In contrast, cDC2s have very limited cross-priming abilities compared to cDC1s, and their roles in anti-tumor immunity are not as well defined.

Furthermore, the data presented in Figure S27 are not persuasive, and it is unclear whether the authors used antigen-specific stimulation or PMA/ionomycin.

In summary, while this manuscript is robust from a chemistry perspective, it contains significant, potentially addressable limitations regarding the rigor of immunogenicity evaluations.

Reviewer #3

(Remarks to the Author)

The authors have done a really good job addressing the prior comments, and I remain supportive of this manuscript. Notably, the chemistry aspects have been significantly improved. The new data concerning shape factor also reveals a

difference between meta, ortho, and para, and further supports the conclusions of the manuscript.

There are a few points remaining however.

Among the previous comments, the authors have not addressed two related points (#5 and #6)

The first is the point around conducting a dose response. As noted before, luciferase “selectively” could be dose dependent, and less apparent at higher doses. Thus, it was recommended to conduct a dose response at doses higher than the original 0.5 mg/kg. Instead, the authors conducted a dose response at lower doses (0.1, 0.2, and 0.3 mg/kg, which is the wrong direction). I sincerely apologize if this confusion was caused by my comment language! But the dose response would still be very important to do at higher doses. I suggest to perform this experiment (e.g. 1.0, 1.5, 2.0, and 3.0 mg/kg) and add the data to a second revision.

The second is the point around the name “Selective Organ-to-Cell Targeting (SSOCT).” No doubt, there is exciting data in this paper with organ tropic delivery. As mentioned in my previous review, the chemistry is very interesting, so the paper can be published in Nature Communications on the basis of that strength alone. But the data in the manuscript does not support a conclusion that the LNPs are both organ selective and cell selective. Even for Fig. 2d, it is likely that at a higher dose (e.g., a dose that would be required for translation to non-human primates or a human clinical study, 1.5 mg/kg – 3.0 mg/kg), this observed cell selectivity may not be apparent. And this is fine. But when readers repeat these studies someday at therapeutically relevant doses, they will likely not see cell selectivity, so it is important to correct the naming and language now, before the paper is published, to write the correct conclusions. Also, the initial submission contained very high background (the free mRNA administered mice have 7-8% TdTomato expression), and this data was deleted in this resubmission, so it is also important to clarify any changes in analysis that generated this revised data and why the new results are definitive and accurate. As mentioned before, endogenous targeting mechanisms may not endow cell selectivity, which is perfectly OK and not a problem for publishing. These new lipids still report a new system for organ selective delivery, which is fantastic.

I also appreciate the inclusion of new Figure S37, which further supports the paper. Please clarify that the axis label (“Tumor Suppression Rate (%)”) is the correct description of the measurement. Seems it could be related to tumor volume or something else, so please check it.

In summary, almost all the concerns have been addressed. I suggest to perform the previously requested dose response at higher doses, as this is important in the context of the major conclusions of the paper, and could affect the naming strategy. But I remain supportive of this paper and think that the conclusions are nearly justified, and will get there with an additional minor study of dose response and clarification of the approach.

Version 2:

Reviewer comments:

Reviewer #1

(Remarks to the Author)

The author has resolved most of the issues, but one minor issue remains to be confirmed. Figure 3b and c show that the dynamic light scattering results indicate consistent particle sizes for mXO10tLNP and mXE14tLNP. However, transmission electron microscopy images reveal that mXE14tLNP appeared smaller, necessitating re-verification of these findings.

Reviewer #2

(Remarks to the Author)

The authors have addressed most of the concerns raised by the reviewers.

Reviewer #3

(Remarks to the Author)

Thank you to the authors for addressing the previous comments.

I can now support the manuscript for publication.

The new dose response results (new Figure S16) are very helpful, and confirm the spleen delivery tropism of the lead LNP. It is confusing why the higher dose of 2 mg/kg reaches 2×10^6 p/s/cm²/sr (new Figure S16), while the prior 10 times lower dose of 0.2 mg/kg also reaches 2×10^6 p/s/cm²/sr or even a little higher (old Figure S15). The authors should make sure these calculations of average radiance are accurate, as the signal should be dose-responsive.

Please also check the data for Figure 2d. It may be better to express the data as bar graph with error bars. It is documented that Ai14 mice have uneven background autofluorescence when comparing between tissues, and that the spleens are uneven compared to other tissues. The only real difference is observed in DC cells between the free mRNA and the mXO10 LNP groups. The violin plot style used here is not appropriate and I suggest changing to bar format for the publication to better show the DC cell contrast (and lack of contrast in other cell types).

The other points were addressed satisfactorily, and I support the paper for acceptance.

The point-to-point responses to the reviewer's comments

Reviewer #1

1. There are already many studies on spleen-targeted LNPs, such as LNP surface modification with antibodies, specific ligand targeting, or using SORT LNP strategies, and achieving spleen targeting by incorporating anionic lipids. In this study, mXO10 has a zeta potential of -38.38 mV, and mXE14 is 2.8 mV, both of which can target the spleen. The paper should explain what mechanism the strategy in this work uses to achieve spleen targeting.

Reply: Thank you very much for your question and for proposing valuable suggestions. As shown, the zeta potential of mXO10 was -38.38 mV, while that of mXE14 was 2.8 mV. Despite their opposite zeta potentials, both LNP exhibited consistent spleen tissue tropism (i.e., a tendency to target the spleen). This may have been attributed to the fact that the LNP used in the initial characterization experiments described in the original manuscript were prepared using manual mixing—a relatively simple method that often led to substantial variability in particle characteristics. Specifically, LNP prepared this way tend to have low encapsulation efficiency and poor particle uniformity, leading to pronounced heterogeneity. This heterogeneity contributes to poor reproducibility of experimental results, with large fluctuations in performance observed across different batches or even at different time points within the same batch. In contrast, the tLNP used in the cell and animal experiments described in the manuscript were prepared using a microfluidic device. Microfluidic preparation allows for precise control over parameters such as flow rate, temperature, and mixing time, ensuring high consistency between batches. This method produces tLNP with uniform particle size and high encapsulation efficiency. Accordingly, we have prepared and characterized both mXO10 tLNP@mRNA and mXE14 tLNP@mRNA using the microfluidic approach. As shown in **Figure 3b**, the average particle size of both mXO10 tLNP@mRNA and mXE14 tLNP@mRNA was approximately 180 nm. We also measured the Zeta potential of tLNP, finding that the surface potential of mXO10 tLNP@mRNA was approximately -15.04 mV, while that of mXE14 tLNP was -17.95 mV. Both tLNP exhibited negative zeta potentials, which was consistent with current findings that negatively charged particles tend to preferentially target the spleen^{1,2}.

Figure 3b. Study on the hydrated particle size and Zeta potential of mXO10 tLNP and mXE14 tLNP.

1. Buschmann, M. D. et al. Nanomaterial Delivery Systems for mRNA Vaccines. *Vaccines* (Basel) **9**, 1 (2021).
2. Žak, M. M. & Zangi, L. Lipid Nanoparticles for Organ-Specific mRNA Therapeutic Delivery. *Pharmaceutics* **13**, 10 (2021).

2. The layout of the paper is not logical. Figure 1 talks about the screening of ionizable lipid libraries and lysosomal escape in DC cells, while Figure 3 discusses LNP preparation and characterization and repeats testing of lysosomal escape in DC cells. Please rearrange the layout for clarity.

Reply: Thank you for your question. We have revised the layout in the resubmitted manuscript to improve the clarity and readability of the paper.

3. mXO10 and mXE14 are considered the most efficient delivery lipids only based on Figure 1. In Figure 1b, many lipids show similar delivery efficiency to mXO10, so why are they not effective?

Reply: We are grateful for the suggestion. In this manuscript, ionizable lipidoids were classified into two main categories based on their carbon chains: one type was synthesized through the reaction of m/o/p-xylylenediamine with 1,2-epoxy carbon chains, represented by mXO10; the other type was synthesized from m/o/p-xylylenediamine and carbon chains containing acrylate groups, represented by mXE14. Many ionizable lipidoids in each category were able to effectively deliver mRNA. However, mXO10 exhibited the highest mRNA delivery efficiency among the 1,2-epoxy carbon chain lipidoids, while mXE14 showed the highest efficiency among the acrylate-containing lipidoids. We selected mXO10 and mXE14 as representative compounds based on their superior delivery performance; this selection did not imply that other ionizable lipidoids were ineffective. Additionally, to manage the workload of subsequent animal experiments, we chose only one ionizable lipidoid from each category for in-depth study.

4. In Figures 2 and 3, data for mXO10 and mXE14 are sometimes compared, and sometimes mXO10 is discussed separately. Please organize the figures and their logic. If mXO10 is the best lipid, then do not repeat tests or comparisons.

Reply: Thank you for your helpful suggestion. We made several adjustments to the layout and logic of certain figures in the manuscript. In this study, ionizable lipidoids were divided into two categories based on their carbon chain structures: one type was synthesized through the reaction of m/o/p-xylylenediamine with 1,2-epoxy carbon chains, represented by mXO10; the other type was synthesized through the reaction of m/o/p-xylylenediamine with carbon chains containing acrylate groups, represented by mXE14. We mainly considered factors such as the layout of the article's figures and the amount of relevant experimental data. In the experiments related to Figures 2 and 3, we compared mXO10 and mXE14. Some parts of the experimental data have been included in the supplementary materials. Please refer to the attached figures. Additionally, a key objective of this research was to investigate whether different types of carbon chains influenced the efficiency of mRNA delivery and whether they affected the performance of positional isomer LNP in mRNA delivery.

5. The DLS results in Figure 3 show that the particle size is around 300 nm. Please aim for a size of around 100 nm and observe the *in vivo* distribution. Additionally, add data on the PDI, thermal

stability, and encapsulation efficiency of the LNP. In Figure 3e, the gel electrophoresis should include an RNA marker, and please refer to <https://doi.org/10.1038/s41541-022-00549-y>.

Reply: Thank you for bringing this issue to our attention. This might have been because the tLNP used in the initial characterization experiments described in the original manuscript were prepared by manual mixing, which resulted in a particle size distribution centered around 300 nm. Because tLNP prepared using this manual method were not uniform—exhibiting significant heterogeneity and poor reproducibility—the tLNP used in the cell and animal experiments in the manuscript were instead prepared using microfluidics. We later used microfluidics to prepare tLNP and conduct their characterization. As shown in the **Figure 3b** and **S19**, the particle size distribution of the microfluidics-prepared mXO10tLNP and mXE14 tLNP was approximately 180 nm. The polydispersity index (PDI) of tLNP was less than 0.2. The figure showed that the particle size of LNP did not change significantly over the course of one week, indicating that the stability of LNP remained good during this period. The mRNA encapsulation efficiency for both mXO10 tLNP and mXE14 tLNP exceeded 70% (**Figure 3d**). As shown in the **Figure S19c**, RNA markers were added to the gel electrophoresis image³.

Figure 3b. Study on the hydrated particle size and Zeta potential of mXO10 tLNP and mXE14 tLNP.

Figure 3d. Study on mRNA encapsulation efficiency of mXO10 tLNP and mXE14 tLNP.

Figure S19. Characterization of mXO10 tLNP and mXE14 tLNP. a and b. Stability study of mXO10 tLNP and mXE14 tLNP. c. Agarose gel electrophoresis to evaluate the resistance of mRNA delivered by mXO10 tLNP (or mXE14 tLNP) to RNase (200 ng/ μ L) and the study of mRNA encapsulation.

3. Voigt, E. A. et al. A self-amplifying RNA vaccine against COVID-19 with long-term room-temperature stability. *NPJ. Vaccines* 7, 136 (2022).

6. The mechanistic exploration in Figure 5 is superficial and does not test specific cellular immune responses. It only looks at general T-cell responses. Please add tests for specific cellular immune responses.

Reply: Thank you very much for your constructive comment. We evaluated the induction of specific cellular responses by mXO10 tLNP@mIDH1 in a mouse glioma model and subsequently detected Th1 and Th2 cells using flow cytometry. The results showed that, compared with the control group, the proportion of Th1 cells in the mXO10 tLNP@mIDH1 group increased significantly, whereas the proportion of Th2 cells did not differ significantly (**Figure S27**). These findings suggest that mXO10 tLNP@mIDH1 immunization induced a Th1-biased cellular immune response.

Figure S27. The induction of specific cellular responses by mXO10 tLNP@mIDH1. Flow cytometry analysis of Th1 and Th2 cell immune subtypes.

7. The paper emphasizes the safety of spleen targeting because there is no accumulation in the liver and kidneys. However, does spleen accumulation increase the burden on the spleen? Please provide

additional evidence that the spleen's normal function is not affected beyond the current safety evaluation.

Reply: Thank you for bringing this issue to our attention. The spleen is a central organ of the immune response. After treating mice with the tumor mRNA vaccine, we evaluated the immune response in the spleen using enzyme-linked immunosorbent assay (ELISA). The results showed that the levels of pro-inflammatory cytokines (IL-6, IL-12, and TNF- α) in the mXO10 tLNP@mIDH1 vaccine group were significantly higher than those in the control group, with the highest levels observed in the group receiving the combination therapy (**Figure 5c** and **S26**). These findings indicated that the spleen's immune function remained active. In addition, we performed hematoxylin and eosin (HE) staining on spleen tissues, and the results showed no signs of tissue damage (**Figure S28**).

Figure 5c. Measurement of cytokine IL-6 in mouse spleens after different treatments using ELISA.

Figure S26. Study on the determination of in vivo cytokines. Measurement of cytokine IL-12 (a) and TNF- α (b) levels in mouse spleens after different treatments using ELISA.

Figure S28. Study on biocompatibility. HE staining of heart, liver, spleen, lung and kidney.

8."Notably, we demonstrated that the delivery of mRNA vaccine (mXO10 tLNP@mIDH1) using meta-tLNP was successful in treating glioma, particularly when combined with Anti-PD-1 therapy, which further improved efficacy and even completely eradicated glioma, while reducing hepatotoxicity and minimizing PEG-lipid-induced allergic reactions." The treatment effect is only observed in mice, please correct this statement.

Reply: Thank you very much for pointing out our mistakes. We have revised this statement in the manuscript.

9."We introduced phenyl rings and hydroxyl functional groups into ionizable lipidoids to improve LNP stability and biocompatibility." Why does adding phenyl rings and hydroxyl groups increase stability? Please explain further in the text.

Reply: I would like to thank you for your question. The incorporation of benzene rings and hydrogen-bonding motifs into the structure of cationic lipidoids in lipid nanoparticles (LNP) significantly enhanced mRNA delivery efficiency through a dual non-covalent interaction mechanism.

(1) π - π stacking interactions: The aromatic phenyl groups in the lipidoids of LNP interact with the aromatic systems of nucleotide bases in mRNA (such as purine and pyrimidine rings) via π - π stacking. This non-covalent interaction is driven by lateral dispersion forces between π -electron clouds and manifests in the following ways: (a) Enhanced loading efficiency: π - π stacking promotes mRNA encapsulation through hydrophobic and aromatic surface complementarity; (b) Stabilization of mRNA-LNP complexes: These reversible stacking interactions synergize with electrostatic forces to enhance complex stability under physiological conditions; (c) Advanced delivery system design strategies: Incorporating aromatic functional groups (such as phenyl and naphthyl) into lipid carriers enables the exploitation of π - π interactions to encapsulate hydrophobic therapeutic molecules (e.g., siRNA and mRNA), resulting in improved loading kinetics and reduced premature release.

(2) Hydrogen bonding interactions: The hydroxyl groups (–OH) on ionizable lipidoids form intermolecular O–H···O hydrogen bonds with the negatively charged phosphate backbone of mRNA. These interactions contribute in the following ways: (a) Strengthened electrostatic binding affinity: Hydrogen bonds act synergistically with the electrostatic interactions of the cationic lipidoids headgroups to form an "electrostatic–hydrogen bond network"; (b) Increased complex structural rigidity: This network reduces dissociation triggered by physiological challenges such as serum proteins and nucleases. (c) Support for rational drug design: The strategic placement of hydrogen bond donors and acceptors on ligands can improve target recognition, including base mimicry and receptor–ligand complementarity^{3,4}.

4. Zhuang, W. R. et al. Applications of π - π stacking interactions in the design of drug-delivery systems. *J. Control. Release.* **294**, 311-326 (2019).

5. Ilhami, F. B., Birhan, Y. S. & Cheng, C. C. Hydrogen-Bonding Interactions from Nucleobase-Decorated Supramolecular Polymer: Synthesis, Self-Assembly and Biomedical Applications. *ACS. Biomater. Sci. Eng.* **10**, 234-254 (2024).

10. "Traditional tumor vaccine is mainly divided into cellular vaccine, peptide vaccine, and DNA vaccine. However, these traditional tumor vaccines suffer from low safety, low protective efficiency, and long production cycles." There are already many mRNA tumor vaccines. Please include the limitations of current mRNA tumor vaccines and explain how this study can address or optimize these issues.

Reply: Thanks for asking such a great question. Despite the safe, flexible, and cost-effective design and production of mRNA tumor vaccines, considerable challenges remain due to their inherent characteristics and technical limitations. Firstly, there are issues with expression stability and persistence. Therapeutic vaccines and protein replacement therapies require prolonged intracellular expression of mRNA drugs *in vivo*, which is limited by the relatively poor stability of mRNA. However, current advancements have improved mRNA stability by optimizing the 5' cap structure, adjusting the length of the 3' poly(A) tail, and modifying regulatory elements in the 5' and 3' untranslated regions. Secondly, RNA has high inherent immunogenicity. *In vitro* synthesized mRNA exhibits high immunogenicity and may induce significant inflammatory responses. Microbial RNA possesses numerous structural and sequential features that distinguish it from host RNA by the human immune system. Two types of pattern recognition receptors (PRRs) in the body can recognize certain conserved components of pathogenic microorganisms, known as pathogen-associated molecular patterns (PAMPs). By activating receptor-specific signal transduction pathways, these receptors induce the expression of pro-inflammatory cytokines and type I interferons (IFN-I), triggering a non-specific antimicrobial response. To address this issue, researchers have found that the immune response to mRNA in humans is mainly related to uridine (partially composed of uracil), and substituting pseudouridine for uracil can reduce the immune system's recognition of mRNA. This modification strategy greatly addresses the immunogenicity of mRNA⁶. Lastly, there is the challenge of *in vivo* delivery of mRNA. Due to its large molecular size and negative charge, mRNA struggles to cross the anionic phospholipid bilayer of the cell membrane, impeding its entry into the cytoplasm for translation into functional proteins and thus limiting its expression *in vivo*. To enable efficient *in vivo* delivery of mRNA, researchers have conducted in-depth studies on various delivery systems, primarily viral and non-viral vectors. Viral

vectors, such as lentiviruses, adeno-associated viruses, and Sendai viruses, can deliver nucleic acids but may be limited by carrier-induced immune responses, affecting their application. Non-viral vectors include liposomes, dendritic cells (DCs), inorganic nanoparticles, cationic cell-penetrating peptides, among others. To date, lipid carriers have emerged as the most effective non-viral vectors for mRNA delivery. However, there is still significant room for improving LNP formulations, such as addressing allergic reactions caused by PEG components and the urgent need for developing delivery systems for cell- or tissue-specific delivery⁷.

6. Pattipeiluhu, R. et al. Anionic Lipid Nanoparticles Preferentially Deliver mRNA to the Hepatic Reticuloendothelial System. *Adv. Mater.* 34, e2201095 (2022).

7. Verbeke, R., Hogan, M. J., Loré, K. & Pardi, N. Innate immune mechanisms of mRNA vaccines. *Immunity* 55, 1993-2005 (2022).

11. "Confocal laser scanning microscopy (CLSM) images showed no significant difference in lysosome escape among meta-, ortho- and para-tLNP (Figure 1c and S7-S9)." How was the conclusion of no significant difference made without statistical analysis? Please perform a quantitative analysis of the images (Figure 1c and S7-S9).

Reply: Thank you for bringing this issue to my attention. We performed semi-quantitative analysis on the confocal images related to tLNP lysosomal escape. The results showed that there was no significant difference in lysosomal escape between mXE14 tLNP and either pXE14 tLNP or oXE14 tLNP, nor between mXO10 tLNP and pXO10 tLNP (**Figure S11**). However, a difference was observed between mXO10 tLNP and oXO10 tLNP, as the cellular uptake rate of oXO10 tLNP was too low to allow a reliable assessment of its lysosomal escape. In summary, these experimental results indicated that there was no significant difference in lysosomal escape among meta-, ortho-, and para-tLNP.

Figure S11. Study on lysosomal escape of mRNA delivered by meta/ortho-/para-tLNP in DC2.4 and GL261 cells. a. Semi-quantitative analysis of lysosomal escape of mXO10/oXO10/pXO10

tLNP in DC2.4 cells (a) and GL261 cells (c). Semi-quantitative analysis of lysosomal escape of mXE14/oXE14/pXE14 tLNP in DC2.4 cells (b) and GL261 cells (d).

12. In Figure 1b, what cells were used to test the transfection efficiency? Please explain this in the figure legend and the text.

Reply: Thank you for bringing this issue to my attention. We used HEK-293 cells in our experiments to assess tLNP transfection efficiency, and have added this information to the figure legend in the manuscript.

13."In contrast, ALC-0315@mLuc showed strong protein expression levels in both the liver and kidneys." The results show expression in the liver and spleen, but not in the kidneys.

Reply: Thank you for pointing out the issue. We have made the corresponding correction to this statement in the manuscript.

14."Therefore, an mRNA delivery system that selectively targets DC cells in the spleen through the SSOCT strategy can not only enhance the therapeutic efficacy of mRNA vaccines but also reduce toxicity to major organs such as the liver and kidneys." Please revise this statement, as the lack of accumulation in the liver and kidneys does not necessarily imply safety.

Reply: Thank for your helpful suggestion. We have made the corresponding correction to this statement in the manuscript.

15.(Figure S13) What method was used for detection, and what dye was used for labeling? Please clarify this in the text and figure legend.

Reply: Thank you for bringing this issue to my attention. We assessed the biodistribution of tLNP in mice by fluorescence imaging, we labeled the mRNA with Cy5 dye, and we have added this information to the text and figure legends of the manuscript.

16.In Figure 2g, why is the fluorescence of mxO10 strongest at 1 hour, and then shows a trend of decreasing, followed by an increase, and then a decrease again? Please check the image and create a fluorescence intensity-time curve. The figure legend is missing.

Reply: Thank you for taking the time to provide such thoughtful and constructive feedback. In Figure 2g, the fluorescence intensity of mXO10@mRNA is highest at 1 hour, followed by a decline, a subsequent rise, and then another decrease. This fluctuation is primarily due to the pulsed expression pattern of mRNA *in vivo*, which allows for a rapid and transient increase in protein levels. mRNA expression not only serves as the foundation for protein synthesis but also plays a critical role in regulating swift and transient protein production through its pulsatile dynamics^{5,6}. Additionally, we have included the fluorescence intensity–time curve and updated the figure legend accordingly, and relevant corrections have been made in the manuscript.

Figure S30. The fluorescence intensity-time curves in Figure 2g.

8. Perry, C. G. et al. Repeated transient mRNA bursts precede increases in transcriptional and mitochondrial proteins during training in human skeletal muscle. *J. Physiol.* 588, 4795-4810 (2010).

9. Golan-Lavi, R. et al. Coordinated Pulses of mRNA and of Protein Translation or Degradation Produce EGF-Induced Protein Bursts. *Cell. Rep.* 18, 3129-3142 (2017).

17. Data in Figure 3f and Figure 1 are repeated; 3g and supplementary Figure S17b are also repeated. The time points on the figure need to be labeled, and the molecular weight (kDa) should be indicated in the figure.

Reply: Thank you for providing your constructive insights on this issue. We made some adjustments to the layout and logical flow of the figures in the manuscript. **Figure 1c** was removed from the manuscript. The content in **Figure 3g** and **S17** was not redundant. **Figure 3g** presented the evaluation of LNP uptake in DC2.4 cells, while **Figure S17** showed a similar experiment conducted in GL261 cells. In both cases, cellular uptake was assessed by imaging after a 4-hour incubation with LNP. We have added the corresponding figure annotations below each figure.

18. Please perform quantitative analysis on Figure 4f.

Reply: We thank you for pointing out this issue. Based on your suggestion, we conducted a quantification of the TUNEL staining in tumor tissues, which corresponds to Figure 4f.

Figure S31. Semi-quantitative analysis of TUNEL staining of tumor tissues in Figure 4f was performed.

19. The y-axis in Figure 5g should represent T-cell statistical analysis, not DC cells.

Reply: Thank you very much for pointing out our mistakes and proposing key amendments. We have made corresponding corrections in our manuscript.

Reviewer #2:

1. Multiple types of DCs reside in the spleen and secondary lymphoid organs. These DC subsets perform different immunological functions. mRNA uptake and protein expression in cDC1s, cDC2s, and pDCs should be determined (instead of using a pan DC marker such as CD11c). The mRNA uptake and expression in macrophages should also be investigated. The mRNA uptake and expression in major lymph nodes should also be determined.

Reply: Thank you for your suggestion. The mXO10 tLNP@tdTomato mRNA was prepared using mXO10 as the ionizable lipidoid to encapsulate tdTomato mRNA. Mice were intravenously injected with mXO10 tLNP@tdTomato mRNA at a dose of 0.2 mg/kg. We measured the expression of tdTomato in various dendritic cell (DC) subtypes—cDC1s, cDC2s, and pDCs—in the spleen and secondary lymphoid organs using flow cytometry. We also measured the expression of tdTomato in macrophages from these tissues using the same method. The results showed that, compared with the control group, the mXO10 tLNP group exhibited no significant differences in tdTomato expression in cDC1s and pDCs, either in the spleen or in secondary lymphoid organs. In contrast, cDC2s showed significantly higher expression levels in both tissues (**Figure S17**). There were also no significant differences in tdTomato expression levels in macrophages from the spleen or secondary lymphoid organs. These results indicated that mXO10 tLNP significantly increased mRNA expression specifically in cDC2s. Additionally, other experimental data in our manuscript showed that there were no differences in the mRNA translation process among the meta-, ortho-, or para-tLNP delivery pathways. Therefore, the level of protein expression served as an indicator of cellular uptake efficiency of mXO10 tLNP. In summary, mXO10 tLNP@mRNA exhibited enhanced mRNA uptake and protein expression in cDC2s.

Figure S17. The development of SSOCT. Study on the targeted expression of mRNA by mXO10 tLNP on DC subsets in spleen and lymph nodes.

2. GL261 cells are typically IDH1 wild-type (IDH1-WT). However, the authors used an mRNA vaccine encoding the IDH1 R132H mutation. This appears to be a major discrepancy which must be clarified.

Reply: We sincerely apologize for the oversight that led to this error. We deeply regretted the confusion, which resulted from our failure to clearly describe the experimental method in the manuscript. Specifically, we did not mention that the GL261 cells used in our study carried the mutant IDH1 gene (IDH1 R132H). In the manuscript, we described the design of a lentiviral plasmid that carried the mutant IDH1 gene, with a sequence for the red fluorescent protein mCherry added downstream of the IDH1 gene sequence to create a fusion protein. Both the plasmid and the lentiviral packaging plasmid were co-transfected into 293T cells to generate the mutant IDH1 gene lentivirus. This lentivirus was then used to infect GL261 cells, and the GL261 cell line carrying the mutant IDH1 gene was selected using puromycin. The following data illustrated our successful generation of the GL261 cell line expressing IDH1 R132H. **Figure S25** showed the expression of the red fluorescent protein mCherry in the GL261 cells, confirming the expression of the IDH1 R132H protein in these cells. Furthermore, we validated the expression of the IDH1 R132H protein in the GL261 cells through Western Blot analysis. For our treatment experiments, we used GL261 cells that had been modified to carry the IDH1 R132H, ensuring that the model more accurately mimicked the pathogenesis of glioma in C57BL/6 mice. We have included the method for generating the GL261 cells with the mutant IDH1 gene (IDH1 R132H) in the experimental section of the manuscript.

Figure S25. Construction of GL261-IDH1 R132H Stable Cell Line. a. Cell morphology and mCherry fluorescence expression: After puromycin selection, the morphology of cells under bright-field microscopy and the expression of mCherry fluorescent protein under fluorescence microscopy were observed. b. Western blot validation: Western blot analysis was performed to verify the expression of the IDH1 gene in GL261 cells.

3. The methods used to characterize T cell responses to the IDH1 R132H encoding mRNA vaccine must be significantly improved. The methodology for the intracellular cytokine staining assays is unclear, making it difficult to assess the immunogenicity of the vaccine.

Reply: Thank for your helpful suggestion. To more accurately evaluate the immunogenicity of the mXO10 tLNP@mIDH1 mRNA vaccine, we treated DC2.4 cells with mXO10 tLNP@mIDH1 and then analyzed the expression of CD80 and CD86 on the cell surface using flow cytometry. This allowed us to assess the activation ability of mXO10 tLNP@mIDH1 on dendritic cells, thereby evaluating the immunogenicity of the LNP. The experimental results showed that, compared with the control group, the proportion of DC cells expressing CD80 and CD86 increased significantly in

the mXO10 tLNP@mIDH1 group, indicating that mXO10 tLNP@mIDH1 effectively activated DC cells and exhibited strong immunogenicity (Figure S32).

Figure S32. Flow cytometric analysis of CD80⁺CD86⁺ in DC2.4 cells after different treatments.

4. Scheme 1 is not informative.

Reply: Thank you for pointing out the issue. Scheme 1 presented the following contents. We synthesized a library of ionizable lipidoids with meta-, ortho-, or para-substitutions using the corresponding phenylenediamine isomers and aliphatic carbon chains. These positionally isomeric ionizable lipidoids were then used, along with phospholipids and cholesterol, to formulate novel three-component lipid nanoparticles (tLNP) for mRNA delivery. Among them, the meta-substituted tLNP (meta-tLNP) demonstrated higher mRNA delivery efficiency compared to ortho-/para-tLNP. Furthermore, all three tLNP variants (meta-, ortho-, and para-) enabled selective mRNA expression in splenic dendritic cells via a sequential selective organ-to-cell targeting (SSOCT) strategy. Notably, the mXO10 tLNP-delivered mIDH1 mRNA vaccine (mXO10 tLNP@mIDH1) showed potent anti-glioma activity, which was further enhanced when combined with anti-PD-1 therapy. We have made corresponding corrections in the manuscript.

5. The Introduction section is too long and repetitive.

Reply: We sincerely appreciate your valuable suggestions. Based on your feedback, we have revised the introduction section of the manuscript to improve its clarity and readability.

6. The rationale for performing uptake studies in GL261 cells is not provided.

Reply: Thank you for the suggestion. GL261 cells were chosen for the cell uptake studies for the following reasons: a. we aimed to determine whether different cell types can effectively take up mXO10 tLNP. The results demonstrated that mXO10 tLNP was efficiently internalized by both dendritic cells and tumor cells. b. The disease model used in this study is a glioma harboring an mutant IDH1 protein (IDH1 R132H). mXO10 tLNP was taken up by GL261 cells, leading to the expression of the mutant IDH1 protein within these glioma cells. Notably, GL261 cells also possess antigen-presenting capabilities. Under certain conditions, they can uptake and process antigens, presenting them to T cells and thereby initiating an anti-tumor immune response.

Reviewer #3:

1. The chemical approach here (summarized in Figure 1a) is a big strength of the manuscript. The direct comparison of meta, ortho, and para is very intriguing. More could be done to further elaborate on this point, because it is a real strength of the manuscript. The compounds synthesized are highly systematic and organized with 3 core and 10 hydrophobic domains of varying length. And the lipid collection and systematic variation is much better executed than similar referenced papers. This chemistry is a strong point of the paper.

Reply: We sincerely appreciate your valuable suggestions. In this study, we synthesized a library of meta-, ortho-, and para-ionizable lipidoids using the corresponding phenylenediamine isomers and aliphatic carbon chains. These positionally isomeric ionizable lipidoids, combined with phospholipids and cholesterol, were used to formulate a series of three-component lipid nanoparticles (meta-/ortho-/para-tLNP) for mRNA delivery. Among them, the meta-tLNP exhibited superior mRNA delivery efficiency compared to the ortho- and para-tLNP. To investigate the underlying mechanism, we conducted experiments such as cellular uptake assays, lysosomal escape analysis, and computational simulation of the structure of ionizable lipidoids. The results revealed no significant differences in lysosomal escape between the three tLNP types (meta-/ortho-/para-tLNP). However, the cellular uptake efficiency of meta-tLNP was significantly higher than that of the ortho- and para-tLNP. To further elucidate the cellular uptake pathways of these tLNP, we employed specific inhibitors to block various endocytic routes. The uptake efficiency of mXO10 tLNP decreased by 63% upon treatment with chloroquine, which alters the pH at the cell membrane and inhibits endocytosis. This suggests that endocytosis is the predominant pathway for cellular uptake of mXO10 tLNP. In contrast, treatment with M β CD reduced the uptake of oXO10 and pXO10 tLNP by 75% and 78%, respectively, indicating that clathrin-dependent endocytosis is the main pathway for their cellular entry. Therefore, the higher cellular uptake efficiency of meta-tLNP contributes to their superior mRNA delivery performance (**Figure S7**).

To understand why positional isomerism leads to differences in cellular uptake, we calculated the P values of the three ionizable lipidoids—mXO10, oXO10 and pXO10—based on the established correlation between lipidoid molecular geometry and mRNA delivery efficiency. The P value, defined as $P = v/(\ell a)$, is a crucial parameter for predicting the nanostructures formed by ionizable lipidoids. Here, a represents the head group area, while v and ℓ denote the tail volume and length, respectively. These physical characteristics greatly influence the self-assembly behavior and function of lipidoid molecules. A P value greater than 1 typically indicates a “small head–large tail” geometry, which favors the formation of inverted structures such as the hexagonal HII phase or inverse spherical nanostructures. The higher the P value, the more readily these structures are formed. Such structures are often associated with enhanced membrane fusion, improved endosomal escape, and more efficient intracellular delivery. Our calculations showed that mXO10 had a P value of 2.1, which was substantially higher than that of oXO10 ($P = 1.6$) and pXO10 ($P = 1.7$). mXO10 with a P value of 2.1 was more likely to assemble into HII phase nanostructures, which provided more favorable conditions for the intracellular delivery of mRNA, which was why the efficiency of cellular uptake of mXO10 tLNP was higher than that of oXO10 tLNP and pXO10 tLNP. Overall, these findings demonstrated that compared with oXO10 and pXO10, mXO10 preferentially formed

HII phase structures, enhancing cellular uptake and leading to superior mRNA delivery by meta-tLNP. Furthermore, these positionally isomeric tLNP exhibited spleen-selective mRNA expression. Given that the spleen is the body's largest immune organ and a central hub for both cellular and humoral immunity, leveraging these tLNP for glioma mRNA vaccine development holds promise for enhancing therapeutic efficacy. This strategy provides a novel avenue for glioma treatment.

Figure S33. Analysis of the reasons for the differences in the delivery efficiency of ionizable lipidoids. a. Representative confocal images showed cellular uptake and endosomal escape of mXO10 tLNP, oXO10 tLNP, and pXO10 tLNP. mRNA was labeled with Cy5 (Red), lysosomes were labeled with Lysotracker (Green), and nuclei were labeled with Hoechst 33342 (Blue). b. The structures of three ionizable lipidoids after MMFF94 force field optimization and the calculation results of P values.

2. The proposed concept here is that meta ionizable amino lipids are superior to para and ortho ionizable amino lipids. Since this is proposed as the key advance, more mechanistic data is needed to prove that definitively. To this point, Scheme 1 introduces the concept but does not explain how or why meta would be superior.

Reply: We are grateful for the suggestion. Ionizable lipidoids are amphiphilic molecules composed of a hydrophilic head and a hydrophobic tail. The molecular packing parameter P is a key physicochemical metric that describes the geometric arrangement and molecular stacking behavior. The P value significantly influences the delivery efficiency of lipid nanoparticles (LNP). In general, lipidoids with relatively high P values tend to adopt an inverted cone-shaped morphology, which facilitates the transformation of the endosomal membrane from its conventional structure into an inverted hexagonal (HII) phase. As a critical cellular structure, the endosomal membrane plays a pivotal role in processes such as intracellular transport and signal transduction. Transitioning into the HII phase induces changes in the intracellular microenvironment that favor the escape of the therapeutic payload from the endosome. Successful endosomal escape enables these payloads to more effectively reach their intracellular targets and perform their intended functions.

Based on an in-depth understanding of the relationship between lipidoid geometry and delivery mRNA efficiency, we calculated the P values of three lipidoids: oXO10, mXO10, and pXO10. The

P value, as an important parameter, provides key insights into predicting the nanostructures that lipidoids may form under specific conditions. Generally, lipidoids with smaller P values (specifically, when the P value is less than 1/3) tend to form spherical structures during aggregation. These spherical structures may exhibit relatively stable physicochemical properties in lipid nanoparticle delivery systems. However, in some cases, they may not be as efficient in delivering payloads as other structures. When the P value falls within the middle range ($1/3 \leq P \leq 1/2$), lipidoids arrange into hexagonal phase (HI phase) nanostructures. When the P value is between 1/2 and 1 ($1/2 < P \leq 1$), planar nanostructures are formed. Hexagonal phase (HI phase) and planar nanostructures have distinct characteristics and mechanisms of action in the delivery process of lipid nanoparticles, and in some cases, they may be more efficient in payload delivery compared to spherical structures. For lipidoids with larger P values (greater than 1), the structural characteristics are typically "small head and large tail." This unique structure increases the likelihood of forming HII phase or inverse spherical nanostructures. The larger the P value, the greater the tendency to form such structures. Both HII phase and inverse spherical nanostructures are closely associated with biological functions, interactions with endosomal membranes, and the delivery of payloads.

According to a formula reported in previous studies in related fields, the P value of lipidoid molecules can be calculated using the equation: $P = v / (\ell a)$. In this equation, a represents the head area of the lipidoid molecule, reflecting the size and shape of its head group, while v and ℓ denote the volume and length of the lipidoid tail, respectively. These physical properties of the tail significantly influence the overall structure and function of the lipidoid molecule. By applying this formula, we can accurately determine the P values of different lipidoids and predict the nanostructures they are likely to form.

After calculation, we found that the P value of mXO10 was 2.1, which was significantly higher than the P values of oXO10 ($P = 1.6$) and pXO10 ($P = 1.7$). Generally, the formation of the HII phase is more likely to induce the rupture of the endosomal membrane in biological systems, which plays a key role in releasing the mRNA payload from the endosome and allowing its entry into the cytoplasm (**Figure 1f** and **1g**). Therefore, mXO10, with a P value of 2.1, was more likely to assemble into HII phase nanostructures, providing more favorable conditions for the intracellular delivery of mRNA. This explained why the efficiency of cellular uptake of mXO10 tLNP was higher than that of oXO10 tLNP and pXO10 tLNP, supporting our consideration of mXO10 in molecular design and demonstrating its potential for future applications.

Figure 1f. 3D structures of mXO10 tLNP, oXO10 tLNP, and pXO10 tLNP lipidoids. Figure 1g. Calculated parameters and P values of mXO10, oXO10, and pXO10 lipidoids.

3. Figure 1b, if meta > ortho/para, then one would expect that the heat map would show this clearly? There appears to be some hits in all three categories. The meta group has the highest overall top hit, but the general hit rate appears to be similar in total hits to ortho and para. So I found these results to be contradictory to the conclusion, and not fully supportive of the hypothesis that meta is superior. Some further investigation and/or explanation may be necessary. Again, the chemistry approach here is awesome – to examine meta/ortho/para systematically, but the current results do not seem to fully support the conclusions.

Reply: Thank you very much for your question. Numerous studies have reported that the efficiency of LNP-mediated mRNA delivery is closely related to the structure of ionizable lipidoids. In our study, the structure of these ionizable lipidoids was divided into two components: the amino head group and the carbon chain tail. Therefore, both the structure of the amino head and the length and type of the carbon chain influence the mRNA delivery efficiency of LNP. In this manuscript, we examined positional isomers (meta, ortho, para) as the amino head groups, combined with carbon chains of varying lengths. As shown in **Figure 1b**, the meta-tLNP group exhibited the highest overall hit rate. Moreover, the top-performing meta-tLNP also showed higher mRNA delivery efficiency than the best-performing ortho- or para tLNP. **Based on both group-level and individual performance, we concluded that the positional isomerism of ionizable lipidoids significantly affects the mRNA delivery efficiency of tLNP—specifically, the meta-tLNP performed better than the ortho- or para-tLNP.** Additionally, we considered the workload of subsequent animal studies and therefore selected ionizable lipidoids with the highest mRNA delivery efficiency—chosen from different positional isomers and two carbon chain categories—as representative candidates for further experiments.

Figure 1b. Heat map showing *in vitro* transfection data after treatment with tLNP made from 30 types of lipidoids encapsulating mLuc.

4. In the introduction and main text and results, the authors claim that endocytosis differences explain why meta lipids are better than ortho/para lipids. More depth in the mechanism is needed to support this conclusion. For example, in the Main Text description and data in Figure 3, some differences in cellular uptake are noted. What is lacking is convincing data to support the mechanism detail. The authors show differences in endocytosis though use of inhibitors, but there is no link to

the LNP chemistry or physical properties. The authors should separate the “chemistry” effects (Is meta really different than ortho/para? HOW does this chemical change directly yield different delivery outcomes conclusively?) and the “physical” effects (Does meta really change the physical properties of LNPs versus ortho/para? HOW do those physical properties change and HOW does that control delivery outcomes?) and potentially “biology” effects (Can cells sense meta LNPs differently than and somehow result in differential delivery outcomes?). These questions are super interesting and important! And the authors have the chemical probes to test it (a very systematic and excellent library of lipids). The current data is unfortunately very light, superficial, and nonspecific, so the conclusions cannot be made. Some definitive and direct links need to be made through additional experiments.

Reply: Thank you very much for your question and proposing important suggestions. We found that the mRNA delivery efficiency of meta-tLNP was higher than that of ortho-tLNP and para-tLNP. We then explored its mechanism of action through experiments such as cell uptake, lysosome escape, and ionizable liposome structure simulation calculation. The results showed a significant difference in the cellular uptake of tLNP between meta-tLNP and ortho-/para-tLNP, indicating that the higher mRNA delivery efficiency of meta-tLNP was primarily due to their enhanced cellular uptake. To understand why positional isomers lead to differences in the cellular uptake of tLNP, we conducted an in-depth study on the relationship between the geometry of ionizable lipidoids and their mRNA delivery efficiency. We calculated the P values of three ionizable lipidoids (mXO10, oXO10 and pXO10). As an important parameter, the P value can provide key clues for us to predict the nanostructures that ionizable lipidoids may form under specific conditions. $P = v/(\ell a)$, a represents the head area of the ionizable lipidoids molecule, which reflects the size and shape of the lipidoids molecule head; v and ℓ represent the tail volume and length of the ionizable lipidoids molecule, respectively. These physical properties of the tail have an important influence on the overall structure and function of the lipidoid molecule. We can calculate the P value of different ionizable lipidoids to predict the nanostructures they may form. When the P value is less than 1/3, these ionizable lipidoids molecules tend to form spherical structures during aggregation. This structure has relatively stable physical and chemical properties, but it is not as conducive to vector delivery as other structures; when $1/3 \leq P \leq 1/2$, ionizable lipidoids will be arranged into hexagonal phase (HI phase) nanostructures; and when $1/2 < P \leq 1$, planar nanostructures will be formed. Hexagonal phase (HI phase) and planar nanostructures have different characteristics and mechanisms of action in the process of lipid nanoparticle delivery. Compared with spherical structures, they may be more conducive to vector delivery in some aspects. When P is greater than 1, this type of ionizable lipidoids has the typical structural characteristics of "small head and large tail". This special structure makes them easier to form HII phase or reverse spherical nanostructures, and the larger the P value, the easier it is to form this structure. The interaction between HII phase and reverse spherical nanostructure and endosomal membrane in biological function effectively improves the delivery efficiency of carriers. After calculation, we found that the P value of mXO10 was 2.1, which was significantly higher than the P value of oXO10 (P = 1.6) and the P value of pXO10 (P = 1.7). mXO10 was more likely to assemble into HII phase nanostructures, which provided more favorable conditions for intracellular delivery of mRNA. This was why the efficiency of cell uptake of mXO10 tLNP was higher than that of oXO10 tLNP and pXO10 tLNP. These experimental results indicated that compared with oXO10 and pXO10, mXO10 was more easily assembled into HII

phase nanostructures, resulting in higher cellular mRNA uptake efficiency of meta-tLNP, thereby making meta-tLNP have higher mRNA delivery efficiency.

Figure S34. a. The confocal images for studying their uptake capacity in DC2.4 cells. b. The semi-quantitative results of the confocal images in (a). c. The molecular structures of the three lipidoids and their P values.

5.mXO10 tLNPs are compared to ALC-0315 LNPs, with descriptions and conclusions in the Main Text summary, as well as data in some instances (e.g., Figure 2b). It is stated in the Main Text conclusions around both efficiency of delivery and tissue tropism (e.g., liver and spleen). Since luciferase bioluminescence is used as a readout, these results may be influenced by the dose. For example, at higher doses, lower expression may become detectable that would change the conclusions around tropism. This is further complicated by the efficacy differences of the LNPs tested. Therefore, it is required to perform a dose response experiment. For example, similar to Figure 2b, the authors can perform a dose response comparing mXO10 tLNPs and ALC-0315 LNPs. This will more accurately benchmark both the delivery efficiency and the tissue protein expression tropism.

Reply: We feel great thanks for your suggestions. In response to your suggestion that dosage might influence targeting specificity, we designed three dosing regimens for mXO10 tLNP@mLuc and ALC-0315 LNP@mLuc: 0.3 mg/kg, 0.2 mg/kg, and 0.1 mg/kg. Bioluminescence imaging was performed 6 hours after administration of the respective doses. The results indicated that, regardless of whether the 0.5 mg/kg dose used in our original manuscript or the lower 0.1 mg/kg dose used in subsequent supplementary experiments, mXO10 tLNP consistently demonstrated spleen-selective mRNA expression. The targeting specificity of tLNP remained unchanged by the variation in dosage (Figure S15).

Figure S15. Dose-dependent expression evaluation of ALC-0315 LNP and mXO10 tLNP. a. The luciferase expression levels after administration of ALC-0315 LNP at doses of 0.1 mg/kg, 0.2 mg/kg, and 0.3 mg/kg. **b.** The luciferase expression levels after administration of mXO10 tLNP at doses of 0.1 mg/kg, 0.2 mg/kg, and 0.3 mg/kg. **c and d.** Semi-quantitative analysis of the luciferase expression levels in **a** and **b**, respectively.

6. The title of Selective Organ-to-Cell Targeting (SSOCT) may need to be changed, unless more convincing data can be generated. From my review, the results are similar to all other active and endogenous targeting LNPs. In other words, there is some organ and some cell pattern but not selective. Active (e.g. antibody conjugated) LNPs have high cell delivery (e.g. to T cells or bone marrow) but very poor organ delivery (most LNPs accumulate in liver). Endogenous (e.g. SORT) LNPs have high organ delivery (e.g. majority of injected dose accumulates in lungs or spleen) but poor cell delivery (transfects most cells in the extrahepatic organ). The authors claim that mXO10 tLNPs are different, but the current data set does not support that conclusion. mXO10 tLNPs are probably using endogenous targeting, and the claim is dendritic cell (DC) specific (e.g. Figure 2d). However, the data shown here (Figure 2d) includes very high background (the free mRNA administered mice have 7-8% TdTomato expression). These results may also be different with lower or higher injected dose. Given that only 4 cell types in the spleen were examined, and that the background signal is super high, one cannot conclude DC cell “selectivity.” The results also show that the tLNPs are not organ selective either (Figure S13). The result here is fine – I just do not think it is realistic for LNPs to be 100% cell selective, nor is there evidence of “SSOCT” in this paper. If the authors want to prove cell selectivity, then they would need to do a dose response. They would also need to quantify delivery using a more accurate method (e.g. the Ai9 or Ai14 mice transfected by Cre recombinase mRNA, or using barcoded mRNA). It would also probably be necessary to quantify delivery to more than 4 cell types (e.g. quantity by using single cell sequencing).

Reply: Thank you again for your valuable suggestions. To investigate the systemic mRNA delivery characteristics of mXO10 tLNP *in vivo*, we encapsulated mRNA encoding firefly luciferase (mLuc) into mXO10 tLNP. Firefly luciferase serves as a reporter protein that can be visualized in live animals using the IVIS imaging system. Following intravenous (tail vein) injection in mice, luciferase expression was monitored over time. A commercial four-component LNP, ALC-0315, was used as a positive control. Interestingly, bioluminescent signals from mice treated with mXO10 tLNP were exclusively detected in the spleen. In contrast, the ALC-0315@mLuc group showed strong luciferase expression in both the liver and spleen. These results indicated that by modifying the structure of ionizable lipidoids, it was possible to achieve organ-selective (e.g., spleen-specific) mRNA expression using LNP. To further examine the cell-targeted mRNA expression of mXO10 tLNP, we used Ai9 mice and administered mXO10 tLNP loaded with Cre mRNA (mCre) at two different doses (0.5 mg/kg and 0.2 mg/kg). Spleen tissues were collected and analyzed via flow cytometry to detect tdTomato-positive cell types, including B cells, T cells, dendritic cells (DC) and NK cells. The experimental results showed that, compared with the control group, a significant increase in tdTomato expression was observed only in DC following treatment with mXO10 tLNP@mCre. Furthermore, the level of tdTomato expression in DC was higher than that observed in the other three cell types (B cells, T cells and NK cells). This suggested that mXO10 tLNP secondarily targeted DC in the spleen to enhance tdTomato expression. Notably, this cell-targeting was consistent across both high- and low-dose groups, indicating that the observed cell-targeting was independent of the administration dose. The SSOCT strategy refers to achieving organ-selective mRNA expression in the first phase, followed by cell-targeted mRNA expression within that organ in the second phase. This enables precise enhancement of mRNA expression in defined cell populations within a target organ. In this case, mXO10 tLNP@mRNA selectively expressed mRNA in the spleen, with no detectable expression in other organs, and subsequently facilitated targeted expression in DC cells within the spleen. Thus, tLNP utilizing the SSOCT approach enabled precise enhancement of mRNA expression in DC populations within the spleen while minimizing off-target expression.

Figure 2d. Study on cell-targeted mRNA expression in the spleen using mXO10 tLNP at a dose of 0.5 mg/kg.

0.2 mg/kg

Figure S16. The development of SSOCT. Study on cell-targeted mRNA expression in the spleen using mXO10 tLNP at a dose of 0.2 mg/kg.

7. One of the amazing and striking results was described in the characterization section. “We also measured the Zeta potential of tLNP, finding that the surface potential of mXO10 tLNP@mRNA was approximately -32.38 mV, while that of mXE14 tLNP was 2.80 mV.” To me, this is a huge difference. It is furthermore amazing that changing the linker can alter surface charge this much. So I wonder if the authors can more precisely pinpoint what changes from meta to ortho/para? This could be the key to the whole study. Does meta change the LNP nanostructure? (Some images are shown in Figure 3c, but are not clear). One could use cryo-TEM and determine if meta lipids organize differently than ortho/para lipids, which could then change the surface nanostructure which could then change protein corona which could then change cellular uptake? And/or does meta change the size and/or surface charge, which could also affect protein corona or other cellular interactions? Better direct proof of what changes (meta versus ortho/para) and how that affects delivery outcomes is needed.

Reply: Thanks for asking such a great question. The charge difference between mXO10 tLNP@mRNA and mXE14 tLNP@mRNA was attributed to the presence of different structural carbon chain tails. These carbon chains contain functional groups with varying charge properties, which influence the overall charge of the tLNP. This study focused on the impact of positional isomers of ionizable lipidoids on the efficiency of LNP-mediated mRNA delivery. Therefore, we compared the delivery efficiency of tLNP composed of different positional isomers, while keeping the carbon chain structure consistent.

Due to an oversight, the two tLNP initially used for characterization experiments in the original manuscript were prepared via manual mixing. Although this method was relatively straightforward, tLNP prepared manually exhibited significant variability. This approach led to low encapsulation efficiency and poor particle uniformity, resulting in pronounced heterogeneity. Consequently, experimental reproducibility was compromised, with performance metrics fluctuating significantly across different batches or even between time points within the same batch. In contrast, the tLNP used in the cell and animal experiments were prepared using microfluidic devices. These devices allow precise control over parameters such as flow rate, temperature, and mixing time, ensuring high batch-to-batch consistency. As a result, the tLNP produced had uniform particle sizes and high encapsulation efficiency. To address the inconsistency, we re-prepared both

types of tLNP using microfluidic methods and conducted new characterization studies. We found that meta-tLNP demonstrated superior mRNA delivery efficiency compared to their ortho- and para-isomer counterparts. The data showed that the hydrated particle sizes of meta-, ortho-, and para-tLNP (with two types of carbon chain tails) were all approximately 180 nm and carried a negative surface charge. This suggested that particle size and charge were not the factors affecting the efficiency of tLNP-mediated mRNA delivery. Further analysis revealed that the geometric structures of mXO10, oXO10, and pXO10 differed significantly. The P value, a critical parameter for predicting nanostructure formation, was used to assess the self-assembly potential of these ionizable lipidoids. The P value is calculated as $P = v / (\ell a)$, where a represents the head group area of the lipidoid molecule, and v and ℓ represent the tail volume and length, respectively. We calculated the P values for each lipidoid: mXO10 had a P value of 2.1, oXO10 had 1.6, and pXO10 had 1.7. A P value of 2.1 suggests that mXO10 was more likely to form hexagonal phase II (HII) nanostructures. The formation of HII phases is known to promote endosomal membrane disruption, facilitating the cytosolic release of mRNA and thereby enhancing the cellular uptake efficiency of mXO10 tLNP.

Figure S35. a. The characterization results of the particle size for the six lipidoids. b. The characterization results of the zeta potential for the six lipidoids. c. The molecular structures of mXO10/oXXO10/pXO10 and the calculation results of their P values.

8. Since the tLNP concept is introduced, whereby no PEG lipid is used, more stability studies are needed. In addition to the reported 3-component LNP diameters being larger than traditional 4-component LNP diameters, one would be concerned with poor stability of PEG free LNPs. Such

LNPs could aggregate in plasma, or adsorb non-specific proteins from plasma, or otherwise have reduced in vivo utility due to poor stability. Studies should be performed and added to characterize tLNP stability in depth with and without added PEG lipid.

Reply: Thank you for your detailed guidance. As you suggested, the absence of PEG components might have affected the stability of LNP. Therefore, we included an evaluation of tLNP stability under both conditions—with and without the addition of PEG lipids—in the revised manuscript. The experimental results showed that tLNP, whether containing PEG lipids or not, exhibited similar particle size trends over a 7-day period, with minimal changes observed. This indicated that tLNP without PEG lipids still maintained good stability. We incorporated phenyl rings and hydroxyl functional groups into ionizable lipidoids to enhance the stability and biocompatibility of LNP. This approach enabled us to construct three-component lipid nanoparticles (tLNP) without PEG-lipids,

Figure S36. The particle size stability of mXO10 tLNP (without PEG component) and mXO10 LNP (with PEG component) after one week of storage at 4°C.

9. The authors claim that mXO10 tLNPs are a superior platform for cancer vaccine therapy (Figure 4). To support this conclusion, the authors should directly compare mXO10 tLNPs to ALC-0315 LNPs in the glioma therapy study (Figure 4). I actually recommend to directly compare meta LNPs to ortho and para LNPs in this therapy study too. That could show superiority and uniqueness of meta lipids. But at a minimum, one would expect benchmarking to ALC-0315 LNPs if this new tLNP platform can be claimed to be better.

Reply: Thank you for your insightful comments and suggestions. We evaluated the therapeutic effects of meta-, ortho-, and para-tLNP-delivered IDH1 mRNA vaccines in a glioma mouse model, using the ALC-0315-delivered IDH1 mRNA vaccine as a control. During the treatment period, we continuously monitored changes in glioma volume in the mice to comprehensively assess the therapeutic efficacy of each vaccine. The experimental results showed that the anti-glioma effect of the mXO10 tLNP@mIDH1 mRNA vaccine was not only greater than that of the oXO10 tLNP@mIDH1 mRNA vaccines, but also superior to that of the pXO10 tLNP@mIDH1 mRNA vaccines. In addition, there was no significant difference in the therapeutic effect between mXO10 tLNP@mIDH1 mRNA vaccine and ALC-0315@mIDH1 mRNA vaccine. These findings indicated that meta-tLNP was a highly effective delivery vehicle for mRNA vaccines and that the mXO10 tLNP@mIDH1 mRNA vaccine represented a promising therapeutic approach for glioma.

Figure S37. Analysis of the therapeutic effects in mice after different treatments. a Tumor growth inhibition rate curves in mice after drug administration. b. Monitoring of body weight changes in mice after drug administration.

10. Some related literature may be cited:

- a. “Paracyclophane-based ionizable lipids for efficient mRNA delivery in vivo. *Journal of Controlled Release*, Volume 376, December 2024, Pages 395-401. This study also looked at positional isomers.
- b. “Iterative Design of Ionizable Lipids for Intramuscular mRNA Delivery” *Journal of the American Chemical Society*, Volume 145, Issue 4, Pages 2294 - 23041 February 2023. This paper examined isomeric variations in ionizable lipids, with some impact for organ-specific delivery.

Reply: Thank you for your careful review. We have already added the corresponding references in the manuscript.

11. LNPs are amorphous particles made via self-assembly. Even though the lipids used are meta/ortho/para, they are not chiral lipids per my understanding. Regardless, it is difficult to fully understand and rationalize how these isomeric and structural differences could affect downstream delivery outcomes. More thought and evidence can be applied to explain the data in this paper. Ultimately, what is the connection? Is it chemical, physical, biological? Protein corona or cell trafficking biology? How could meta/ortho/para mediate cell specific delivery? There is a lot of missing space between chemical synthesis and cell specific delivery. Explaining with data how exactly meta lipids mediate these changes would be super interesting and an amazing advance for the field!

Reply: Thank you for this helpful suggestion. Our study found that the mRNA delivery efficiency of meta-tLNP was higher than that of ortho-tLNP and para-tLNP, as reflected by its superior cellular uptake. Specifically, the cellular uptake efficiency of meta-tLNP exceeded that of both ortho- and para-tLNP. To explore the underlying reasons for this difference, we characterized the meta-, ortho-, and para-tLNP. The results showed that all three formulations had a similar hydrated particle size of approximately 180 nm and carried a negative surface charge, suggesting that particle size and surface charge were not responsible for the differences in cellular uptake.

Previous literature has reported a close relationship between the geometry of ionizable lipidoids and their mRNA delivery efficiency. To investigate this, we calculated the P values of the three ionizable lipidoids: oXO10, mXO10, and pXO10. The P value, defined as $P = v / (\ell a)$, is a key parameter used to predict the nanostructures formed by ionizable lipidoids under specific

conditions. Here, a represents the head group area of the lipidoid molecule (reflecting the size and shape of the head), while v and ℓ denote the volume and length of the lipidoid tails, respectively.

According to established structural models:

When $P < 1/3$, lipidoids tend to form spherical structures with stable physicochemical properties but relatively low delivery efficiency.

When $1/3 \leq P \leq 1/2$, lipidoids may form inverse hexagonal (HI) phase nanostructures. When $1/2 < P \leq 1$, they are likely to form planar nanostructures. Compared with spherical structures, HI phase and planar nanostructures may be more favorable for certain aspects of vector delivery during lipid nanoparticle-mediated transport.

When $P > 1$, lipidoids typically exhibit a "small head–large tail" architecture and are more likely to form inverse hexagonal (HII) or inverse spherical structures. The larger the P value, the greater the tendency to form such structures. These HII and inverse spherical nanostructures are believed to interact more effectively with endosomal membranes, thereby enhancing intracellular delivery efficiency.

Our calculations revealed that the P value of mXO10 was 2.1, while oXO10 and pXO10 had P values of 1.6 and 1.7, respectively. The high P value of mXO10 indicated a greater tendency to form HII phase nanostructures, which were known to facilitate endosomal escape and intracellular mRNA delivery. This structural propensity likely accounted for the higher cellular uptake and delivery efficiency observed with mXO10 tLNP compared to oXO10 and pXO10 tLNP. In summary, our findings suggested that among the positional isomers studied, mXO10 was more prone to assembling into HII phase nanostructures, resulting in superior mRNA delivery efficiency and higher cellular uptake than its ortho and para counterparts.

Figure S38. a. The characterization results of the particle size for mXO10/oXO10/pXO10 tLNP. b. The semi-quantitative results of the confocal images of the uptake of mXO10/oXO10/pXO10 tLNP in DC2.4 cells. c. The confocal images of lysosomal escape for mXO10/oXO10/pXO10 tLNP. d.

3D structures of mXO10, oXO10, and pXO10 lipidoids. e. Calculated parameters and P values of mXO10, oXO10, and pXO10 lipidoids.

Guanjun Deng, Ph.D
School of Pharmaceutical Sciences (Shenzhen)
Sun Yat-sen University
Shenzhen 518107, P.R. China
E-mail: denggj3@mail.sysu.edu.cn

September 3, 2025

Dear Editor, Dear reviewers

Thank you for your letter dated June 9. We were pleased to know that our work was rated as potentially acceptable for publication in Journal, subject to adequate revision. We thank the reviewers for time and effort that they have put into reviewing the previous version of the manuscript. Their suggestions have enabled us to improve our work. Based on the instructions provided in your letter, we uploaded the file of the revised manuscript.

Appended to this letter is our point-by-point response to the comments raised by the reviewers. The comments are reproduced and our responses are given directly afterward in a different color (Red).

We would like also to thank you for allowing us to resubmit a revised copy of the manuscript.

We hope that the revised manuscript is accepted for publication in *Nature Communications*. Should you have any questions, please contact us without hesitate.

Sincerely,

Guanjun Deng

The point-to-point responses to the reviewer's comments

Reviewer #1:

1. The current optimized LNP preparation protocol has improved particle homogeneity, while the zeta potential of mXE14 changed dramatically from +2.8 mV to -17.95 mV. Such a significant alteration would inevitably invalidate all previously acquired experimental data. Therefore, all related experiments should be re-evaluated using the updated LNP.

Reply: Thank you for this helpful suggestion. In response to your suggestion, we have re-characterized the two types of tLNP prepared by microfluidics in terms of their pKa and TEM images. Transmission electron microscopy (TEM) images showed that both mXO10 tLNP@mRNA and mXE14 tLNP@mRNA had a dense, spherical structure, with nanoparticle sizes around 170 nm (**Figure 3c**). The pKa of tLNP was determined using 6-(p-toluidino)-2-naphthalenesulfonic acid (TNS) assays, indicating pKa values of 4.45 and 5.70 for mXO10 tLNP@mRNA and mXE14 tLNP@mRNA, respectively (**Figure 3e**). These experiments were conducted to more accurately characterize the physicochemical properties of the microfluidics-prepared tLNP and to ensure consistency with those used in animal experiments. This approach minimizes potential deviations in experimental results that could arise from discrepancies in tLNP physicochemical properties. Additionally, the previous version of the manuscript has been updated with data on the particle size, PDI, zeta potential, encapsulation efficiency, gel electrophoresis assessment of encapsulation, and particle size stability of the tLNP prepared by microfluidics.

Figure 3c. Representative TEM images of mXO10 tLNP and mXE14 tLNP. Scale bar: 0.2 μm .

Figure 3e. Study on the pKa values of mXO10 tLNP and mXE14 tLNP.

2. The evaluation of antigen-specific immune cells represents a critical criterion for assessing cancer vaccine efficacy. These data should be presented in the main text rather than supplementary materials. Although the author provided data on Th1 and Th2, there are still some doubts. What are the markers selected for detecting TH1 and TH2? What is the specific operation method? What does it mean that mXO10 tLNP@mRNA 1/3 samples induced the production of IFN- γ ⁺ IL-4⁺ double positive? What is the logic of the flow gate? In addition, ELISPOT is also an important method for detecting specific cellular immunity.

Reply: We feel great thanks for your suggestions. We have presented the data on the assessment of antigen-specific immune cells in the main text.

(1) To detect Th1 and Th2 cells, we selected CD3 and CD4 as surface markers, and IL-4 and IFN- γ as intracellular markers. The specific experimental procedures were as follows:

1) Isolate splenocytes from the spleen and prepare a single-cell suspension.

2) Stimulation and blocking with 10% FBS-RPMI 1640 medium containing IDH1 R132H-specific antigenic peptides and BFA for 4 hours. After stimulation and blocking, centrifuge at 500 xg for 5 minutes at 4°C, and discard the supernatant.

3) CD16/32 receptor blockade: Add 100 μ L of cell staining buffer containing 1 μ g of CD16/32 Fc receptor blocker, and incubate on ice in the dark for 10 minutes.

4) Surface staining: Add 1 μ L of CD3-FITC and 1 μ L of CD4-APC to each tube, and incubate on ice in the dark for 15 minutes.

5) Fixation and permeabilization: Use the Cyto-Fast™ fix/perm buffer set for fixation and permeabilization.

6) Intracellular cytokine staining: Resuspend the cells in 100 μ L of 1 \times Cyto-Fast™ perm/wash solution and add 1 μ L of IL-4-PE and 1 μ L of IFN- γ -BV421. Mix well and incubate at room temperature in the dark for 20 minutes.

7) Washing and resuspension: Wash the cells with 500 μ L of 1 \times Cyto-Fast™ perm/wash solution. Centrifuge at 400 xg for 5 minutes at 4°C, discard the supernatant, and resuspend the cells in 300 μ L of cell staining buffer.

Flow cytometry was used to analyze Th1 and Th2 immune cell subtypes after XO10 tLNP@mIDH1 treatment. The results showed that, compared with the control group, the proportion of Th1 cells in the mXO10 tLNP@mIDH1 group increased significantly, whereas the proportion of Th2 cells did not differ significantly. These results demonstrated that mXO10 tLNP@mIDH1 immunization induced a Th1-biased cellular immune response (**Figure 5g and S33**).

Figure S33. Flow cytometry analysis of Th1 and Th2 cell immune subtypes after XO10 tLNP@mIDH1 treatment.

Figure 5g. The induction of specific cellular responses by mXO10 tLNP@mIDH1. Flow cytometric statistical analysis of Th1 and Th2 immune cell subtypes.

(2) The IFN- γ ⁺IL-4⁺ double-positive cells induced in 1/3 of the mXO10 tLNP@mRNA samples may have resulted from the intrinsic ability of mXO10 tLNP to simultaneously activate Th1 and Th2 immune responses.

(3) Flow cytometry gating strategy (Figure S34):

1) Lymphocyte gating: First, gate on the lymphocyte population based on FSC-A (forward scatter area) vs. SSC-A (side scatter area).

2) Single cell gating: Next, use FSC-A vs. FSC-H (forward scatter height) to exclude doublets and gate on single cells.

3) T cell gating: Then, gate on CD3⁺ cells to identify T cells.

4) Th cell gating: Further, within the T cell population, gate on CD4⁺ cells to identify Th cells.

5) Th1 and Th2 subsets: Finally, based on IFN- γ ⁺ and IL-4⁺ expression, respectively gate on Th1 cells and Th2 cells.

Figure S34. Flow cytometry gating strategy

(4) In accordance with your suggestion, we assessed the levels of the cytokines IFN- γ and IL-4 secreted using the ELISpot assay to further support the aforementioned results. As shown in the **Figure S30**, the mXO10 tLNP group elicited significantly higher levels of IFN- γ compared to the

control group, while no significant difference was observed in IL-4 levels. These findings suggest that mXO10 tLNP primarily induces a Th1-type immune response during the treatment of glioblastoma, thereby combating the tumor.

Figure S30. Determination of cytokines IFN- γ and IL-4. a and b. The secretion levels of the cytokines IFN- γ and IL-4 following treatment with mXO10 tLNP were evaluated using the ELISpot assay. c. The secretion of IFN- γ and IL-4 cytokines after treatment with mXO10 tLNP was quantitatively assessed using the ELISpot assay.

3. For all fluorescence microscopy images and histopathological sections presented in both the main text and supplementary materials, the scale bar must be supplemented.

Reply: Thank you for your careful review. We have added scale bars to the corresponding images.

Reviewer #2:

1. One major concern is the analysis of cytotoxic T lymphocytes (Figure 5G). This analysis was conducted using non-specific stimulation with PMA and ionomycin rather than antigen-specific stimulation. As a result, no conclusions can be drawn about the immunogenicity of the vaccine based on these studies. It is essential for the authors to adopt an antigen-specific approach to measure mutant IDH-specific CD8⁺ T cells. Typically, vaccine-elicited CD8⁺ T cells should recognize an epitope corresponding to the IDH1 R132H mutation. Therefore, the ICS assay should utilize peptides containing this mutated epitope instead of relying on PMA and ionomycin. The immunogenicity of the new mRNA-LNP vaccines cannot be accurately assessed without measuring antigen-specific CD8⁺ T cell responses, ideally using commercial LNPs as a benchmark.

Reply: We are very grateful for your suggestions. In response, we re-evaluated the antigen-specific response of CD8⁺ T cells using IDH1 R132H-specific peptides and included ALC-0315 as a positive control. Accordingly, we used a peptide containing the IDH1 R132H mutant epitope in this assay to assess vaccine-elicited CD8⁺ T cells, rather than relying on PMA and ionomycin. As shown in the Figure S29 and S35. The number of CD3⁺CD8⁺IFN- γ ⁺ T cells in the mXO10 tLNP@mIDH1 mRNA

vaccine group was significantly higher than that in the control group. This result indicated that mXO10 tLNP@mIDH1 effectively stimulated the production of IDH1 R132H-specific CD3⁺CD8⁺IFN- γ ⁺ T cells. This finding further substantiated the therapeutic potential of mXO10 tLNP.

Figure S35. Flow cytometry analysis of IDH1 R132H-specific CD3⁺CD8⁺IFN- γ ⁺ T cells in the mXO10 tLNP@mIDH1 group.

Figure S29. Determination of IDH1 R132H-specific CD3⁺CD8⁺IFN- γ ⁺ T cells in the mXO10 tLNP@mIDH1 group. Flow cytometric statistical analysis of IDH1 R132H-specific CD3⁺CD8⁺IFN- γ ⁺ T cells

2. Additionally, the significance of the finding stating, “These results indicated that mXO10 tLNP significantly increased mRNA expression specifically in cDC2s,” is unclear in the context of anti-tumor immunity. cDC1s are critical for mediating CD8⁺ T cell anti-tumor responses and play pivotal roles in priming CD4 T cell responses. In contrast, cDC2s have very limited cross-priming abilities compared to cDC1s, and their roles in anti-tumor immunity are not as well defined.

Reply: Thank you very much for your question. While cDC1 cells are adept at cross-presenting

antigens and activating CD8⁺ T cells, thus serving as the "main force" in anti-tumor immunity, cDC2 cells may exhibit more abundant expression and have the capability to present antigens to CD4⁺ T cells. This could potentially facilitate anti-tumor immunity indirectly by activating CD4⁺ T cells. Additionally, the results from tumor tissue CD4⁺/CD8⁺ T cell immunohistochemistry can further support the activation of cDC2 cells by our material.

3. Furthermore, the data presented in Figure S27 are not persuasive, and it is unclear whether the authors used antigen-specific stimulation or PMA/ionomycin.

Reply: Thank you again for your valuable suggestions. We apologize for the oversight in our initial experiments, where we used PMA/ionomycin to stimulate cytokine production, which does not allow us to accurately assess the antigen-specific immune response elicited by our vaccine. Following your valuable suggestion, we have re-evaluated the immune response using IDH1 R132H-specific antigenic peptides. As shown in the **Figure 5g and S33**, stimulation with IDH1 R132H-specific antigenic peptides predominantly induces a Th1-type immune response. This finding underscores the vaccine's capacity to elicit a robust and specific Th1-mediated response, which is crucial for effective immune activation against the targeted antigen. We are grateful for your guidance and will continue to refine our experimental design to ensure more accurate and meaningful outcomes in future studies. We have updated the corresponding figure in the manuscript.

Figure S33. The induction of specific cellular responses by mXO10 tLNP@mIDH1. Flow cytometry analysis of Th1 and Th2 cell immune subtypes.

Figure 5g. The induction of specific cellular responses by mXO10 tLNP@mIDH1. Flow cytometric

statistical analysis of Th1 and Th2 immune cell subtypes.

Reviewer #3 (Remarks to the Author):

1. Among the previous comments, the authors have not addressed two related points (#5 and #6). The first is the point around conducting a dose response. As noted before, luciferase “selectively” could be dose dependent, and less apparent at higher doses. Thus, it was recommended to conduct a dose response at doses higher than the original 0.5 mg/kg. Instead, the authors conducted a dose response at lower doses (0.1, 0.2, and 0.3 mg/kg, which is the wrong direction). I sincerely apologize if this confusion was caused by my comment language! But the dose response would still be very important to do at higher doses. I suggest to perform this experiment (e.g. 1.0, 1.5, 2.0, and 3.0 mg/kg) and add the data to a second revision.

Reply: Thank you very much for your question and proposing important suggestions. In response to your suggestion, we conducted in vivo mRNA expression experiments at higher doses, and the results showed that even at 2 mg/kg, mXO10 tLNP maintained its spleen-selective mRNA expression properties (Figure S16). Unfortunately, we were unable to conduct in vivo mRNA expression studies at the 3 mg/kg dose. During the experiments, we observed that the mXO10 tLNP@mRNA nanoparticles underwent fragmentation at this dose, which made it impossible to prepare a 3 mg/kg formulation. Therefore, we did not proceed with the evaluation of organ-selective mRNA expression at 3 mg/kg. We sincerely hope for your understanding regarding this limitation.

Figure S16. Dose-dependent mRNA expression evaluation of ALC-0315 LNP and mXO10 tLNP. a. mRNA expression of mXO10 tLNP at different doses (mRNA expression levels after administration of mXO10 tLNP at doses of 1 mg/kg, 1.5 mg/kg, and 2 mg/kg). b. mRNA expression of ALC-0315 LNP at different doses (mRNA expression levels after administration of ALC-0315 LNP at doses of 1 mg/kg, 1.5 mg/kg, and 2 mg/kg). c and d. Semi-quantitative analysis of the

expression levels in a and b, respectively.

2. The second is the point around the name “Selective Organ-to-Cell Targeting (SSOCT).” No doubt, there is exciting data in this paper with organ tropic delivery. As mentioned in my previous review, the chemistry is very interesting, so the paper can be published in Nature Communications on the basis of that strength alone. But the data in the manuscript does not support a conclusion that the LNPs are both organ selective and cell selective. Even for Fig. 2d, it is likely that at a higher dose (e.g., a dose that would be required for translation to non-human primates or a human clinical study, 1.5 mg/kg – 3.0 mg/kg), this observed cell selectivity may not be apparent. And this is fine. But when readers repeat these studies someday at therapeutically relevant doses, they will likely not see cell selectivity, so it is important to correct the naming and language now, before the paper is published, to write the correct conclusions. Also, the initial submission contained very high background (the free mRNA administered mice have 7-8% TdTomato expression), and this data was deleted in this resubmission, so it is also important to clarify any changes in analysis that generated this revised data and why the new results are definitive and accurate. As mentioned before, endogenous targeting mechanisms may not endow cell selectivity, which is perfectly OK and not a problem for publishing. These new lipids still report a new system for organ selective delivery, which is fantastic.

Reply: Thank you very much for your question and for proposing valuable suggestions. In response to your feedback, we revised the definition of SSOCT. For the high background tdTomato expression in the initial submission, which was due to the relatively low sensitivity of tdTomato mRNA detection in regular C57 mice and potential background signal interference, we employed Ai9 mice in the revised manuscript. The Ai9 model, with its highly sensitive transgenic reporter system, enabled accurate detection of low-level tdTomato mRNA activity and reduced background noise, making it ideal for assessing the activation of mRNA expression in different cell types in mouse spleens by mXO10 tLNP.

3. I also appreciate the inclusion of new Figure S37, which further supports the paper. Please clarify that the axis label (“Tumor Suppression Rate (%)”) is the correct description of the measurement. Seems it could be related to tumor volume or something else, so please check it.

Reply: Thank you for your careful review. In the supplementary figure S37, the axis label "Tumor Suppression Rate (%)" is indeed related to tumor volume. The assessment is based on the tumor volume on day 8 after implantation, which is set as the baseline value of 1. The tumor volumes on subsequent days (days 11, 13, and 18) are compared to the baseline volume on day 8 to determine whether there is any inhibition in tumor volume growth. This further supports the therapeutic efficacy of our material.

Figure S37. Analysis of the therapeutic effects in mice after different treatments. a Tumor growth inhibition rate curves in mice after drug administration. b. Monitoring of body weight changes in mice after drug administration.

4. In summary, almost all the concerns have been addressed. I suggest to perform the previously requested dose response at higher doses, as this is important in the context of the major conclusions of the paper, and could affect the naming strategy. But I remain supportive of this paper and think that the conclusions are nearly justified, and will get there with an additional minor study of dose response and clarification of the approach.

Reply: Thank you very much for your suggestion. Before proceeding to higher doses, we performed reaction analyses at 0.1 mg/kg, 0.2 mg/kg, 0.3 mg/kg, 1 mg/kg, 1.5 mg/kg, and 2 mg/kg. The results showed that mXO10 tLNP maintained its spleen-selective mRNA expression properties across doses ranging from 0.1 to 2 mg/kg.

Guanjun Deng, Ph.D
School of Pharmaceutical Sciences (Shenzhen)
Sun Yat-sen University
Shenzhen 518107, P.R. China
E-mail: denggj3@mail.sysu.edu.cn

October 11, 2025

Dear Editor, Dear reviewers

Thank you for your letter dated September 27. We were pleased to know that our work was rated as potentially acceptable for publication in Journal, subject to adequate revision. We thank the reviewers for time and effort that they have put into reviewing the previous version of the manuscript. Their suggestions have enabled us to improve our work. Based on the instructions provided in your letter, we uploaded the file of the revised manuscript.

Appended to this letter is our point-by-point response to the comments raised by the reviewers. The comments are reproduced and our responses are given directly afterward in a different color (Red).

We would like also to thank you for allowing us to resubmit a revised copy of the manuscript.

We hope that the revised manuscript is accepted for publication in *Nature Communications*. Should you have any questions, please contact us without hesitate.

Sincerely,

Guanjun Deng

The point-to-point responses to the reviewer's comments

Reviewer #1:

1. The author has resolved most of the issues, but one minor issue remains to be confirmed. Figure 3b and c show that the dynamic light scattering results indicate consistent particle sizes for mXO10tLNP and mXE14tLNP. However, transmission electron microscopy images reveal that mXE14tLNP appeared smaller, necessitating re-verification of these findings.

Reply: Thank you for your careful review. We sincerely apologize for this oversight. Upon re-examination, although both TEM images of the tLNP were labeled 20 μm , the actual scale-bar lengths differed due to variations in magnification, which exaggerated the apparent size difference. We have now harmonized the magnification and scale bars. Thank you again for bringing this to our attention.

Figure 4c. Representative TEM images of mXO10 tLNP and mXE14 tLNP.

Reviewer #3:

1. The new dose response results (new Figure S16) are very helpful, and confirm the spleen delivery tropism of the lead LNP. It is confusing why the higher dose of 2 mg/kg reaches 2×10^6 p/s/cm²/sr (new Figure S16), while the prior 10 times lower dose of 0.2 mg/kg also reaches 2×10^6 p/s/cm²/sr or even a little higher (old Figure S15). The authors should make sure these calculations of average radiance are accurate, as the signal should be dose-responsive.

Reply: We are very grateful for your suggestions. We regret the confusion caused by this inadvertent error. After re-examination, the data were misrepresented due to a scaling error during

figure preparation; the plot has now been corrected and redrawn. We are grateful for your astute feedback.

Figure S16. Dose-dependent mRNA expression evaluation of ALC-0315 LNP and mXO10 tLNP. a. mRNA expression of mXO10 tLNP at different doses (mRNA expression levels after administration of mXO10 tLNP at doses of 1 mg/kg, 1.5 mg/kg, and 2 mg/kg). b. mRNA expression of ALC-0315 LNP at different doses (mRNA expression levels after administration of ALC-0315 LNP at doses of 1 mg/kg, 1.5 mg/kg, and 2 mg/kg). c and d. Semi-quantitative analysis of the expression levels in a and b, respectively.

2. Please also check the data for Figure 2d. It may be better to express the data as bar graph with error bars. It is documented that Ai14 mice have uneven background autofluorescence when comparing between tissues, and that the spleens are uneven compared to other tissues. The only real difference is observed in DC cells between the free mRNA and the mXO10 LNP groups. The violin plot style used here is not appropriate and I suggest changing to bar format for the publication to better show the DC cell contrast (and lack of contrast in other cell types).

Reply: Thank you very much for your question. Following your suggestion, we have revised the presentation of Figure 2d into a bar format.

Figure 3d. Study on targeted delivery of mRNA to cells in the spleen using mXO10 tLNP at a dose of 0.5 mg/kg.